# The longitudinal dynamics and natural history of clonal haematopoiesis

Margarete A. Fabre[1,2,3,10], José Guilherme de Almeida[4,10], Edoardo Fiorillo[5], Emily Mitchell[1,2,3], Aristi Damaskou[2,3], Justyna Rak[2,3], Valeria Orrù[5], Michele Marongiu[5], Michael Spencer Chapman[1,2,3], M. S. Vijayabaskar[2,3], Joanna Baxter[6], Claire Hardy[1], Federico Abascal[1], Nicholas Williams[1,2], Jyoti Nangalia[1,2,3], Iñigo Martincorena[1], Peter J. Campbell[1,2], Eoin F. McKinney[7], Francesco Cucca[5,8], Moritz Gerstung[4,9] ✉ & George S. Vassiliou[1,2,3] ✉

Clonal expansions driven by somatic mutations become pervasive across human tissues with age, including in the haematopoietic system, where the phenomenon is termed clonal haematopoiesis[1–4]. The understanding of how and when clonal haematopoiesis develops, the factors that govern its behaviour, how it interacts with ageing and how these variables relate to malignant progression remains limited[5,6]. Here we track 697 clonal haematopoiesis clones from 385 individuals 55 years of age or older over a median of 13 years. We find that 92.4% of clones expanded at a stable exponential rate over the study period, with different mutations driving substantially different growth rates, ranging from 5% (*DNMT3A* and *TP53*) to more than 50% per year (*SRSF2*[P95H]). Growth rates of clones with the same mutation differed by approximately ±5% per year, proportionally affecting slow drivers more substantially. By combining our time-series data with phylogenetic analysis of 1,731 whole-genome sequences of haematopoietic colonies from 7 individuals from an older age group, we reveal distinct patterns of lifelong clonal behaviour. *DNMT3A*-mutant clones preferentially expanded early in life and displayed slower growth in old age, in the context of an increasingly competitive oligoclonal landscape. By contrast, splicing gene mutations drove expansion only later in life, whereas *TET2*-mutant clones emerged across all ages. Finally, we show that mutations driving faster clonal growth carry a higher risk of malignant progression. Our findings characterize the lifelong natural history of clonal haematopoiesis and give fundamental insights into the interactions between somatic mutation, ageing and clonal selection.

Human haematopoiesis produces hundreds of billions of specialized blood cells every day, through a hierarchy of progressively more differentiated and numerous cells originating from a pool of long-lived haematopoietic stem cells (HSCs). Haematopoiesis remains highly efficient for decades, but is inevitably challenged by the erosive effects of ageing[7–9] and the inexorable acquisition of somatic DNA mutations[10]. Mutations that augment HSC fitness can drive clonal expansion of a mutant HSC and its progeny, a phenomenon known as clonal haematopoiesis[1–4]. Clonal haematopoiesis becomes ubiquitous with advancing age and is associated with an increased risk of myeloid leukaemias and some non-haematological diseases[1,2,4,5,11,12].

The observation that clonal haematopoiesis-associated mutations affect a restricted set of genes that are also frequently mutated in leukaemia[1–4]—most commonly those involved in epigenetic regulation (*DNMT3A*, *TET2* and *ASXL1*), splicing (*SF3B1* and *SRSF2*) and apoptosis

(*TP53* and *PPM1D*)—implies that these mutations inherently confer fitness to HSCs. In fact, recent evolutionary models assume that each specific mutation carries a fixed fitness advantage, and find that this largely explains the relative proportions and clonal sizes of clonal haematopoiesis driven by different mutations[13]. However, several observations suggest that non-mutational factors are also influential. For example, a handful of clonal haematopoiesis cases studied at two time points propose that clones driven by the same or similar mutations can behave differently between individuals[12,14]. Also, the relative prevalence of different clonal haematopoiesis-driver gene mutations changes significantly depending on context; for example, in aplastic anaemia, clonal haematopoiesis is commonly driven by mutations that enhance immune evasion[15–18], whereas genotoxic stress favours clones with mutations in DNA damage genes[19–21]. Furthermore, factors

[1]Wellcome Sanger Institute, Wellcome Genome Campus, Cambridge, UK. [2]Wellcome-MRC Cambridge Stem Cell Institute, University of Cambridge, Cambridge, UK. [3]Department of Haematology, University of Cambridge, Cambridge, UK. [4]European Molecular Biology Laboratory, European Bioinformatics Institute EMBL-EBI, Wellcome Genome Campus, Cambridge, UK. [5]Istituto di Ricerca Genetica e Biomedica, Consiglio Nazionale delle Ricerche, Lanusei, Italy. [6]Cambridge Blood and Stem Cell Biobank, Department of Haematology, University of Cambridge, Cambridge, UK. [7]Cambridge Institute of Therapeutic Immunology & Infectious Disease, University of Cambridge, Cambridge, UK. [8]Dipartimento di Scienze Biomediche, Università degli Studi di Sassari, Sassari, Italy. [9]Division of AI in Oncology, German Cancer Research Centre DKFZ, Heidelberg, Germany. [10]These authors contributed equally: Margarete A. Fabre, José Guilherme de Almeida. ✉e-mail: moritz.gerstung@dkfz.de; gsv20@cam.ac.uk

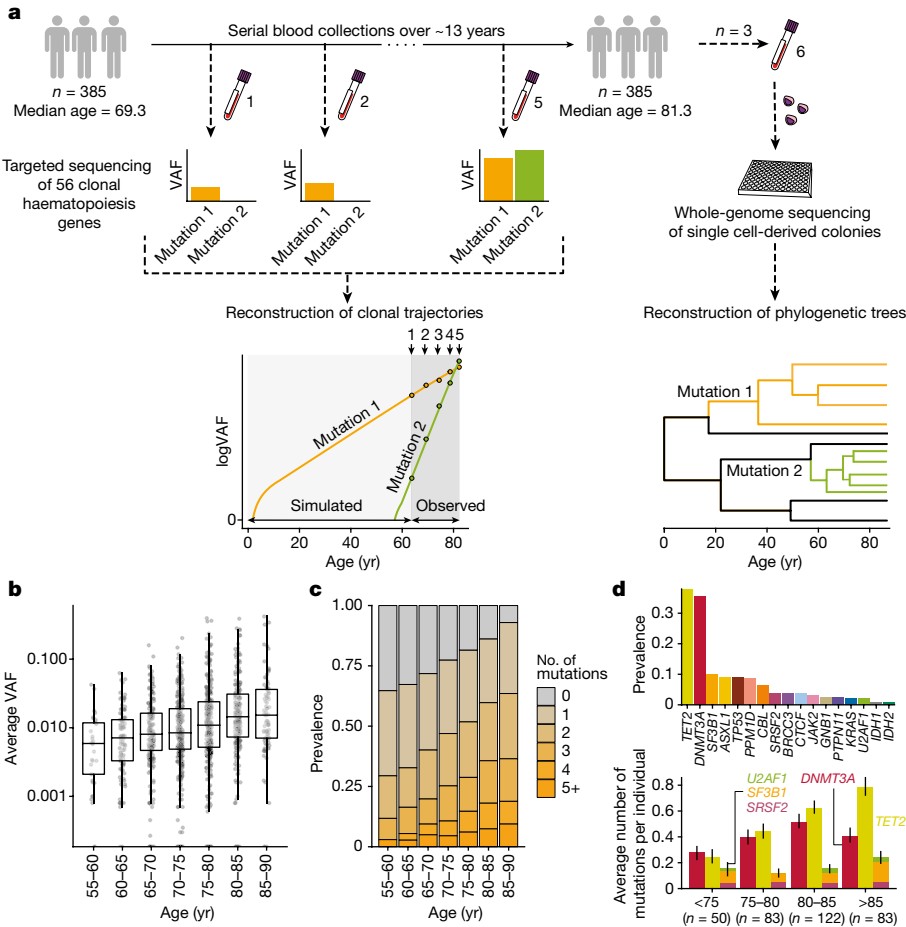

**Fig. 1 | Experimental workflow and clonal haematopoiesis mutation characteristics. a**, Study outline: 1,593 blood DNA samples were obtained from 385 elderly individuals sampled 2–5 times (median 4) over 3.2–16 years (median 12.9) and sequenced for mutations in 56 clonal haematopoiesis genes. Measured VAFs were used to fit observed clonal trajectories and extrapolate the clonal dynamics prior to the period of observation. Additional blood samples from 3 selected individuals were used to generate 288 (that is, 3 × 96) whole-genome-sequenced single cell-derived colonies for phylogeny reconstructions. **b**, Age distribution of average VAF per individual (*n* = 1,258 VAF measurements). The boxes represent the 25th, 50th (median) and 75th percentiles of the data; the whiskers represent the lowest (or highest) datum within 1 interquartile range from the 25th (or 75th) percentile. **c**, Age-stratified prevalence of the number of mutations per individual. **d**, Prevalence of mutations in driver genes. Top, absolute prevalence in the cohort. Bottom, average number of mutations per individual in *DNMT3A*, *TET2* and splicing genes (*SF3B1*, *SRSF2* and *U2AF1*) at different ages, with error bars representing bootstrap 90% confidence intervals.

such as inflammation[22] and heritable genetic variation[23–25] can affect the emergence of clonal haematopoiesis.

A major limitation to our understanding of the determinants of clonal haematopoiesis behaviour and fate up to now has been its reliance on cross-sectional studies capturing clonal haematopoiesis at single time points. Here, by tracking blood cell clones over long periods in a large cohort, and by reconstructing haematopoietic phylogenies, we uncover the lifelong dynamics and natural history of clonal haematopoiesis.

## Mutational landscape of clonal haematopoiesis

We analysed 1,593 blood DNA samples from 385 adults aged 55–93 years at the time of entry into the SardiNIA longitudinal study[26]. The participants, who had no history of haematological malignancy, were sampled up to 5 times (median 4) over 3.2–16 years (median 12.9 years) (Fig. 1a, Extended Data Fig. 1a–c). We performed deep sequencing (mean 1,065× coverage) of 56 genes associated with clonal haematopoiesis and haematological malignancy (Supplementary Table 1) and identified somatic mutations in 52 of these genes (Supplementary Table 2). Using the dNdScv algorithm, an implementation of d*N*/d*S* (the ratio of the number of nonsynonymous substitutions per non-synonymous site to

the number of synonymous substitutions per synonymous site) that corrects for trinucleotide mutation rates, sequence composition and variable mutation rates across genes, we identified positive selection of missense and/or truncating variants in 17 of these genes[27] (d*N*/d*S* > 1 with *q* < 0.1) (Extended Data Figs. 1d–e, 2, Supplementary Table 3). We focussed on these genes for further analysis.

At least one somatic non-synonymous mutation was identified in 305 of 385 individuals (79.2%), with clonal haematopoiesis prevalence, average clone size and number of mutations per individual increasing with advancing age, and clonal haematopoiesis was identified in more than 90% of those aged 85 years or older (Fig. 1b, c). Mutations were most common in epigenetic regulator genes *TET2* and *DNMT3A*, and also frequent in *ASXL1*, *TP53*, *PPM1D* and spliceosome genes (Fig. 1d, top). Notably, in this elderly cohort, advancing age affected the prevalence of different driver mutations in a gene-dependent manner (Fig. 1d, bottom). In particular, the prevalence of *DNMT3A* mutations showed no significant relationship with age overall (*P* = 0.12 for a binomial regression of gene prevalence versus age, controlling for sex). By contrast, clones with *TET2* mutations showed a consistent increase with age, averaging at 6.8% per year (*P* = 0.00037), as did those with mutations in splicing genes (*U2AF1*, *SRSF2* and *SF3B1*), whose prevalence increased by 5.4%

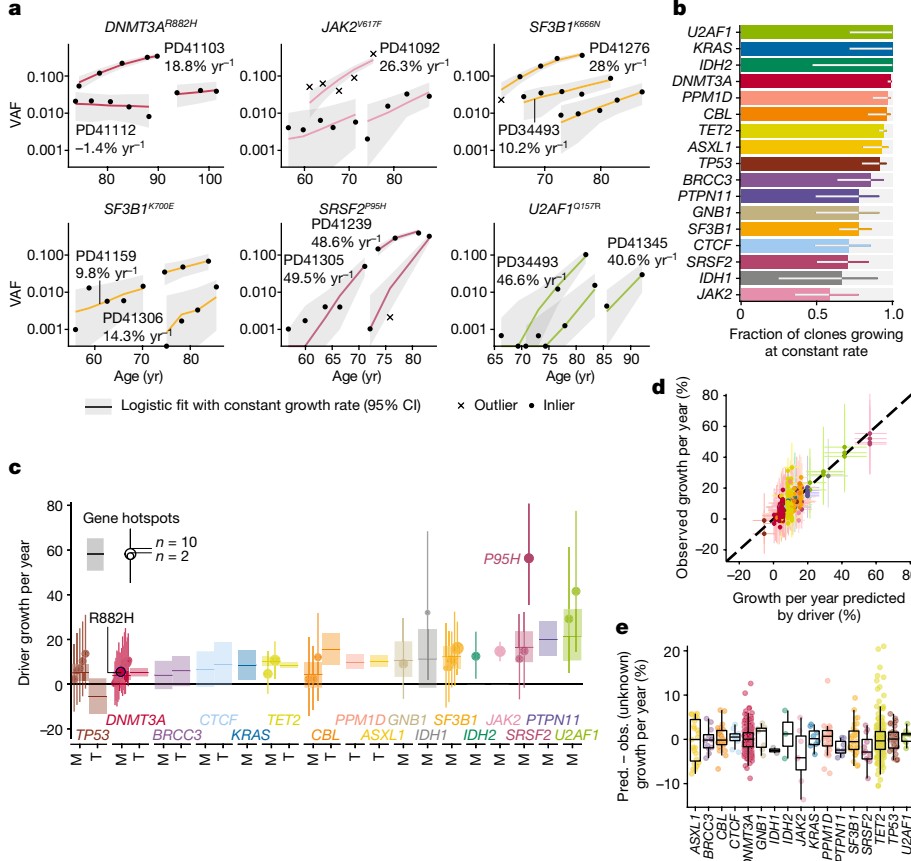

**Fig. 2 | The longitudinal dynamics of clonal haematopoiesis in older age.**
**a**, Examples of fitted exponential growth of clones with mutations at six common hotspots. Points represent observed data, coloured lines represent estimated VAF trajectories and grey bands represent the 90% highest posterior density interval (HPDI). Each data point is represented by a dot if it conforms to our model of fixed-rate exponential growth and by a cross otherwise (outlier, defined as tail probability <2.5%). **b**, Proportion of clonal trajectories showing fixed-rate growth—that is, those with no outlying data-points as defined in **a**. Bars represent the proportion and error bars represent the 90% beta-distributed confidence interval. **c**, Annual clonal growth associated with different driver mutations, for both genes and specific sites. For gene-wise growth, truncating (T) and missense (M) mutations are modelled separately for genes where both are enriched. Sites are modelled separately to genes if mutated recurrently within our cohort. Point estimates for growth and 90%

HPDI are represented for each site (dot and line, respectively, with dot size proportional to recurrence) and each gene (horizontal line and rectangle, respectively). **d**, Relationship between clonal growth predicted by the identity of the driver mutation and actual observed growth (points), with 90% HPDI represented by vertical and horizontal lines, respectively. $n = 633$ clones. **e**, Distribution of the unknown-cause effect for different genes. Each point represents a single clone and box plots represent the distribution of these effects for each gene. The value of unknown-cause growth is positive for clones growing faster than expected by the identity of the driver mutation, and negative for clones growing slower than expected ($n = 633$ clones). The boxes represent the 25th, 50th (median) and 75th percentiles of the data; the whiskers represent the lowest (or highest) datum within 1 interquartile range from the 25th (or 75th) percentile. Pred., predicted; obs., observed. CI, confidence interval.

per year ($P = 0.025$). These changes in driver prevalence with age could not have resulted from exclusion of individuals with haematological malignancies, as the incidence of these in the complete SardiNIA cohort was only 0.28% (22 out of 7,816), the majority of which were lymphoid.

## Most clones expand steadily in older age

To investigate clonal behaviour over time, we used serial variant allele fraction (VAF) measurements—the fraction of sequencing reads reporting a mutation—as a surrogate for clone size, and fitted a saturating (logistic) exponential curve with a constant growth rate over time to each clonal trajectory. Such logistic growth behaviour is supported by simulations of evolutionary dynamics using Wright–Fisher models with constant fitness[28] (Extended Data Fig. 3a, b). Remarkably, by assessing the fit between serial VAF measurements and the trajectories inferred by our model, we find that the great majority of clones (92.4%) expanded at a constant exponential rate over the study period (Fig. 2a, b, Extended Data Fig. 3c). The predominance of fixed-rate growth was particularly marked for genes such as *DNMT3A* and *TET2*, for which 99% and 94.3% of

clones, respectively, grew steadily over time. Nevertheless, some clones behaved unpredictably, with proportions varying by mutant gene. Most notable were *JAK2*[V617F]-mutant clones, which showed irregular growth trajectories, with only 58% displaying stable growth. The likelihood of mutant clones displaying non-constant growth at older age was not affected either by the number of mutations in the same individual or by the number of available serial samples (Extended Data Fig. 3d, e).

We further assessed the consistency of clonal trajectories by testing our ability to predict future clonal growth. Using additional prospectively-obtained blood samples from 11 individuals, we compared observed versus predicted VAFs (Extended Data Fig. 3f–h, Supplementary Table 4) and found good concordance (mean absolute error 3.5%), corroborating our model and providing further evidence that fixed-rate growth of clones is the norm in old age.

## Determinants of clonal growth rate

To delineate the factors that determine each clone's growth rate, our logistic regression model fits the following contributions of the

driver mutation: (1) mutated gene; (2) specific amino acid change (in recurrently mutated sites), and (3) mutation type (truncating versus non-truncating) (Supplementary Table 5). An additional component in our model, measuring variation not captured by (1)–(3), was also used and termed 'unknown-cause growth' (Extended Data Fig. 3i).

We found that clones bearing mutations in different genes expanded at different rates, with mutations affecting *DNMT3A* and *TP53* displaying the slowest average annual growth rates of approximately 5% per year (Fig. 2c, Supplementary Table 6). Clones with mutations in the other most common driver genes (*TET2*, *ASXL1*, *PPM1D* and *SF3B1*), expanded at roughly twice this rate, that is, about 10% per year. The most rapidly expanding clones were those carrying mutations in *SRSF2*, *PTPN11* and *U2AF1*, which grew at 15–20% per yr on average. The only specific mutation displaying distinctive behaviour was *SRSF2*[P95H], which was associated with significantly faster expansion compared with other *SRSF2* mutations. By contrast, all other hotspot mutations drove growth at rates similar to mutations elsewhere in the same gene, including commonly mutated sites such as *DNMT3A*[R882], *SF3B1*[K666N] and *SF3B1*[K700E].

For most genes, truncating and missense mutations drove similar rates of growth, including *TET2* and *DNMT3A*, in keeping with the similar functional consequences of these two types of mutation in these genes[29,30]. Exceptions were (1) *TP53*, for which clones with missense mutations expanded by 10% per year (90% confidence interval [3–18%]) faster than truncating mutations (which usually did not expand or even contracted), consistent with the reported strong dominant-negative effect of missense mutations in this gene[31], and (2) *CBL*, for which clones with missense mutations grew 11% per year (90% confidence interval [3–19%]) slower than truncating mutations (Fig. 2c, Extended Data Fig. 3j, Supplementary Table 6).

To quantify the impact of factors other than driver mutations, we compared the observed growth rate of each clone with that predicted by the mutation (Fig. 2d). In Fig. 2d, vertical spread represents variability in growth rate between clones with the same driver mutation. On average, this growth of unknown cause contributed approximately ±5% per year to clonal expansion (Fig. 2e). Consequently, for fast-growing clones, including those associated with *SRSF2*[P95H] or mutant *U2AF1*, this effect was proportionately small and there was relatively little inter-individual variability in growth rate. By contrast, the effect on slow drivers such as *DNMT3A* was more substantial, with some clones growing twice as rapidly as predicted by the mutation, and others showing negligible expansion. Clones harbouring *JAK2*[V617F] mutations were an exception as they displayed an unusually high degree of inter-individual variability in relation to average growth rate (Fig. 2d, e, Extended Data Fig. 4a). In view of the well-described heritable contribution to myeloproliferative neoplasm (MPN) susceptibility[23,24], we tested whether *JAK2*[V617F]-mutant clones grew more quickly in individuals carrying MPN risk alleles, but found no such relationship (Extended Data Fig. 4b, Supplementary Table 7).

The more general observation that certain individuals harboured more mutations in the same gene than would be expected by chance (Extended Data Fig. 4c) suggests that non-mutation factors influencing clonal growth are both individual- and gene-specific. We found no evidence that these non-mutation factors include either sex or smoking history and that initial clone size made only a small contribution, whereas age was a significant factor specifically for *TET2*-mutant clones, which grew faster in older individuals (Spearman's rho = 0.31; sum of squared rank differences ($S$) = $1.15 \times 10^6$; $n = 216$ *TET2* clones; adjusted $P = 2 \times 10^{-6}$) (Extended Data Fig. 4d–g).

## Lifelong natural history of clonal haematopoiesis

To compare the longitudinal clonal behaviours we observed in older age with lifelong clonal dynamics, we began by deriving and whole-genome sequencing (WGS) 96 haematopoietic colonies, each originating from a single stem or progenitor cell and expanded in vitro to a clone of hundreds to thousands of cells, from each of three individuals with

splicing gene mutations (Fig. 3a–c, Extended Data Fig. 5), particularly as previous reports suggested a sharp increase in prevalence of these driver mutations late in life[3]. We constructed phylogenetic trees using somatic mutations as lineage-tracing barcodes and, since HSCs accumulate mutations at a near constant rate, we used phylogenetic branch lengths to time the onset of clonal expansions ('clades')[32–37]. In PD41276, the phylogeny was dominated by an *SF3B1*[K666N]-mutant clone, beginning between 23–47 years of age, with only a single *SF3B1*-wild-type colony, consistent with a near-complete clonal sweep (Fig. 3a). In PD34493, *SF3B1*[K666N] was acquired before the age of 35 years, whereas *U2AF1*[Q157R] initiated clonal growth later (41–61 years of age) in a previously expanded clade lacking recognizable drivers (Fig. 3b). Notably, an additional apparently driverless expansion—a phenomenon that has been recognized to occur in old age[2,36]—and three further such expansions in PD41305 were observed in this individual (Fig. 3b), (Fig. 3c). In PD41305, since the *SRSF2*[P95H] mutation was present in only one colony, we could time its acquisition only to the broad interval between 13 years of age and the age of sampling (73 years of age).

We next used the timing and density of clonal branchings (also known as 'coalescences') to reconstruct the entire growth trajectories of expanded clades using phylodynamic principles[33,38,39] (Fig. 3d–h). This revealed that the three clades with identified drivers (*SF3B1*[K666N] and *U2AF1*[Q157R] in PD34493, and *SF3B1*[K666N] in PD41276), expanded (Fig. 3d–f) at calculated rates similar to those observed in our time-series VAF measurements during older age (Fig. 3i, left). Of note, *SF3B1*[K666N] was associated with a substantially different growth rate in PD41276, where it expanded at 28% per year according to serial VAFs (29% per year by phylodynamic estimate), versus 10% per year in PD34493 (17% per year by phylodynamics) (Fig. 3i). Reasons for this difference are unclear, but it is notable that the faster-growing clone had antecedent Y loss (Fig. 3a), an aberration seen in clades from all three individuals and associated with only modest clonal expansion when isolated (Fig. 3a–c). Of note, clones without known drivers began to expand within the first two decades of life and grew over their lifetimes at rates similar to clones with known drivers (14–32% per year) (Fig. 3g, h, Extended Data Fig. 6).

## Many clones decelerate before older age

As the phylodynamic reconstruction of a clone goes back to its inception, we investigated whether clonal growth dynamics during earlier life deviate from the stable growth observed during older age. To corroborate observations from the three individuals depicted in Fig. 3, we conducted additional phylodynamic analyses of trees derived from 1,461 whole-genome-sequenced single cell-derived colonies from another four individuals 75–81 years of age from the study by Mitchell et al.[36]. This revealed that, in many instances, the reconstructed effective population size ($N_{eff}$) of any individual clone grew more slowly towards the sampling date and before it saturated the HSC compartment (Fig. 4a, b, Extended Data Fig. 7a–c). This characteristic deceleration was quantified by fitting a biphasic exponential growth model to early and late parts of the trajectories (Fig. 4c). In most cases, extrapolating early growth (a consistent estimator of the fitness advantage of a clone in Wright–Fisher simulations; Extended Data Figs. 7d, 8) led to substantial overestimations of clade size (median 35×; Fig. 4d, Extended Data Fig. 7e).

We used our longitudinal cohort to orthogonally test the lifelong stability of clonal growth by extrapolating the observed (fitted) trajectory of each clone backwards in time to infer the age at clonal onset. To account for stochastic drift, which can lead to faster growth of small clones, and the finite carrying capacity of the HSC population, which naturally limits or slows large clones, we derived and used an approximation to a Wright–Fisher process (Extended Data Fig. 4a, b). Whereas estimates of age at clonal onset agreed with phylogenetic estimates for the fast-growing splice factor mutations (Fig. 3i), for many other clones, constant lifelong growth at the rate we observed during old age would

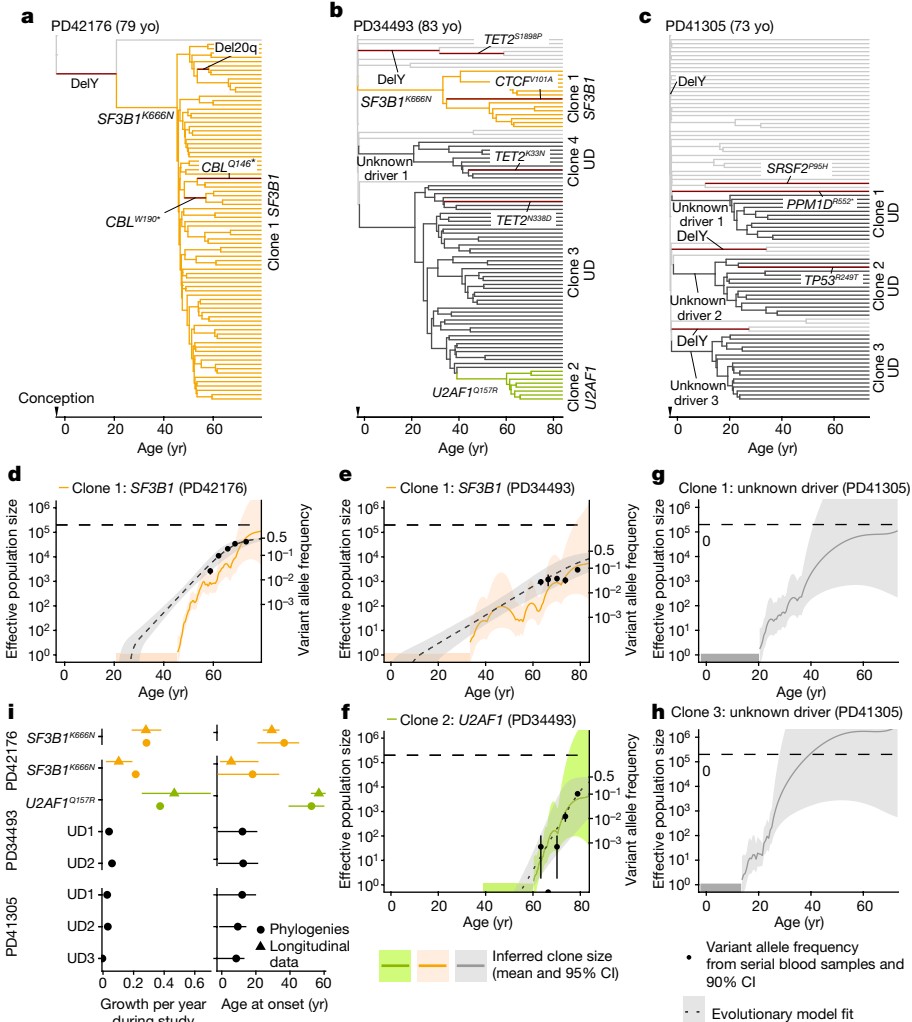

**Fig. 3 | Haematopoietic phylogenetic trees. a–c**, Haematopoietic phylogenies of participants PD41276 (**a**), PD34493 (**b**) and PD41305 (**c**). Each tree tip is a single cell-derived colony and tips with shared mutations coalesce to an ancestral branch, from which all colonies in such a 'clade' arose. Branch lengths are proportional to the number of somatic mutations, which accumulate linearly with age except before birth, at which point approximately 55 mutations have been acquired[36]. Branches containing known driver mutations or chromosomal aberrations are annotated. Clonal expansions are coloured: $SF3B1^{K666N}$-mutant expansions in orange, $U2AF1^{Q157R}$-mutant expansions in green, and expansions without identified drivers (unknown driver (UD)) in black. **d–h**, Growth trajectories of each clonal expansion, as determined by phylogenies (effective population size ($N_{eff}$) estimated using phylodynamic methods) and time-series data (using serial VAF measurements and modelled historical growth, as illustrated in Fig. 2, if available). $SF3B1$-mutant expansions for PD42176 (**d**) and PD34493 (**e**), $U2AF1$-mutant expansions for PD34493 (**f**), and unknown driver expansions clone 1 (**g**) and clone 3 (**h**) for PD41305. Phylogeny-derived age at clone onset range is represented as a horizontal coloured bar on the x-axis, with the limits of the bar corresponding to the age range of the phylogeny branch along which the corresponding driver mutation was acquired. **i**, Comparison of the ages at onset (right) and growth rate during the study period (left) derived from phylogenetic trees and longitudinal data. For the age at onset and growth rates derived from longitudinal data, the intervals represent the 90% HPDI; age at onset intervals derived from phylogenies represent the age limits defined by phylogenetic branching patterns. For annual growth estimates using phylogenies, intervals represent the standard error. yo, years old.

be too slow to explain the observed VAFs (Fig. 4e–g), suggesting that clonal expansion was faster in earlier life. These observations reveal that, at least for some clones and genes, the dynamics observed in later life are not representative of those that prevail earlier.

We then assessed the minimum lifetime rate at which clones must have grown in order to reach the observed VAFs in our longitudinal data—hereafter termed 'historical growth'—by restricting fits and solutions to growth rates that would place the age of clonal onset within individuals' lifetimes (Fig. 4h, Supplementary Table 8). Expectedly, this minimal historical growth rate was typically higher than the growth rate observed during the study period (that is, in older age; Fig. 4i, Extended Data Fig. 7f). Moreover, the fold changes between historical and observed growth rates derived from longitudinal data were qualitatively in good agreement with the fold changes between late

growth and expected growth (the latter assuming growth is constant through life and carrying capacity is fixed) derived from phylodynamic data (Fig. 4c, i, Extended Data Fig. 7f). Thus it emerges that many clones grew more rapidly early in life compared with the rate in old age.

## Driver genes and lifelong clonal growth

The effect of deceleration was most marked for clones bearing mutations in *DNMT3A*, *BRCC3* and *TP53*, whose early growth was at least twice as fast as that measured during old age (Fig. 4i, j). Conversely, we observed almost no deceleration of fast-growing clones harbouring *U2AF1*, *SRSF2^P95H*, *PTPN11* or *IDH1* mutations (Fig. 4i, j). It is particularly notable that the *TET2*-mutant clones were much less susceptible to deceleration than *DNMT3A*-mutant clones (Fig. 4i, j). This is consistent

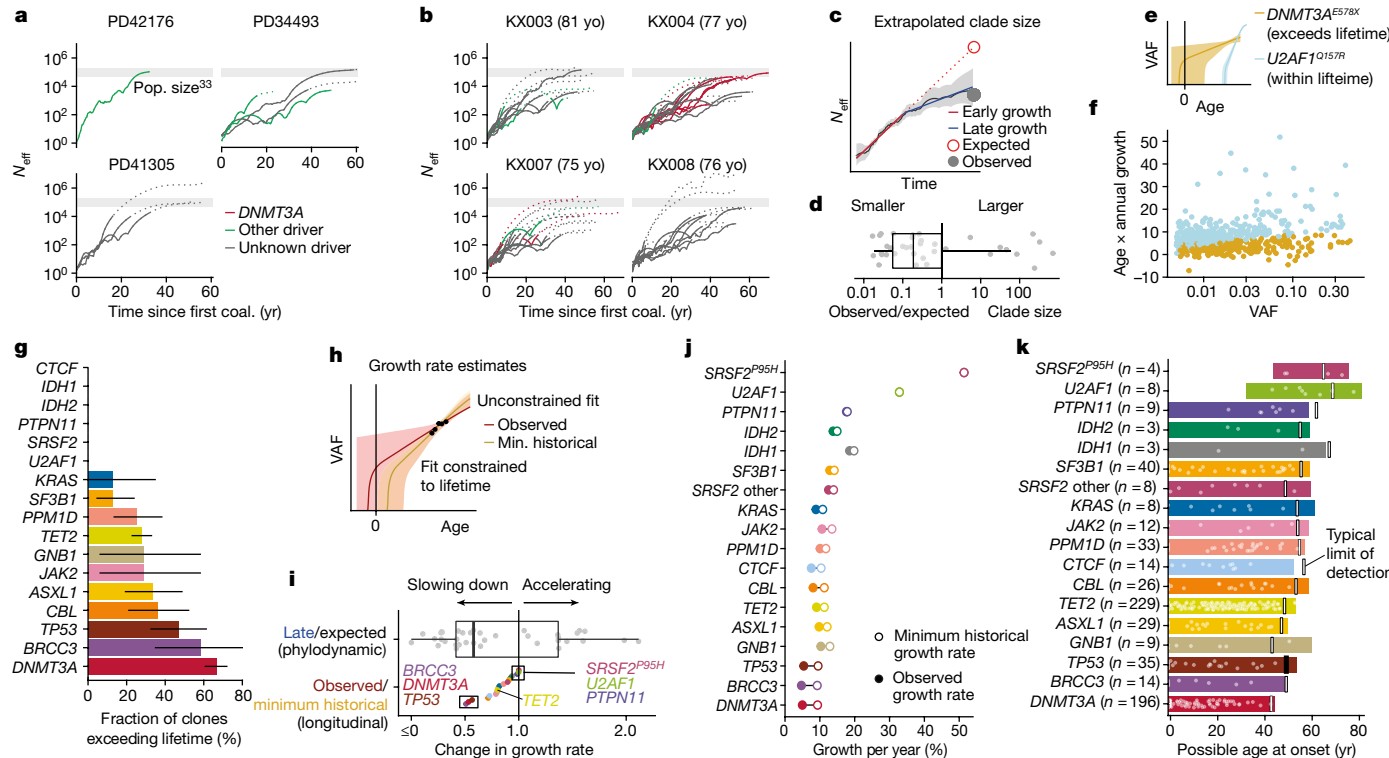

**Fig. 4 | Evidence for clonal deceleration from single-cell phylogenies and longitudinal data. a, b**, Effective population size ($N_{eff}$) trajectories inferred from single-cell phylogenies in this paper (**a**) and in Mitchell et al.[36], using previously determined HSC population size estimates[33] (**b**). Dotted lines represent parts of the trajectory with high variance ($\log(\mathrm{var}(N_{eff})) > 5$). Coal., coalescence. **c**, Representation of biphasic fit to $N_{eff}$ estimates and extrapolation from early growth (observed clone size is calculated as the clonal fraction in the phylogeny scaled by an $N_{eff}$ of 200,000 HSCs × yr; comparison with 1,000,000 HSC × yr in Extended Data Fig. 7e). **d**, Ratio of observed to expected (extrapolated from early growth) clone size from phylogenies ($n = 37$ expanded clones detected in haematopoietic phylogenies). **e**, Representation of extrapolated trajectories derived from longitudinal data, assuming stable lifelong growth at the same fixed rate we observed during older age; some projections are not feasible (that is, they exceed lifetime, with onset pre-conception). **f**, Relationship between age

and observed growth rate of clones and VAF (longitudinal data; light blue represents clones with projected onset within lifetime and golden represents those exceeding lifetime). **g**, Quantification of unfeasible clones (exceeding lifetime) per gene (longitudinal data, $n = 633$). Intervals represent the beta-distributed 90% confidence interval. **h**, Representation of the calculation of minimum (min.) historical growth. **i**, Ratio of observed to historical (longitudinal data) and late to expected (phylogenetic data) growth ($n = 37$ clones detected in phylogenies (top); $n = 633$ in longitudinal data (bottom)). **j**, Differences between the median observed and historical growth per year for each gene. **k**, Projected ages at onset for all clones, assuming stable lifelong growth at the same fixed rate we observed during older age. Boxes in **d**, **i**, represent the 25th, 50th (median) and 75th percentiles of the data; the whiskers represent the lowest (or highest) datum within 1 interquartile range from the 25th (or 75th) percentile.

with the observation that the prevalence of *TET2*-mutant clonal haematopoiesis is higher at older ages and eventually exceeds that of *DNMT3A*-mutant clonal haematopoiesis, which is more prevalent at younger ages (Fig. 1d). A declining relative advantage of *DNMT3A* mutations in older age was also suggested by the much lower proportion of *DNMT3A*-mutant clones reaching detectable limits during our study period compared with clones bearing mutations in other genes ('incipient clones') (Extended Data Fig. 9a).

To derive representative ranges for age at clone onset for each driver gene, we capped individual estimates at conception, thus avoiding estimates that projected beyond individuals' lifetimes (Fig. 4k, Extended Data Fig. 9b, c). We also validated this method using simulations and confirmed that these ranges are not affected by changes in $N_{eff}$ or generation time (Extended Data Fig. 9d, e). We estimated that the average age latency between clone foundation and detection in peripheral blood at VAF ≥ 0.2% (Supplementary Note 1) was 30 years across all clones, with considerable variability between mutant genes, ranging from 38 years for *DNMT3A*-mutant clones to 12 years for *U2AF1*-mutant clones. Most drivers were projected to initiate expansions of clones throughout life, compatible with the notion that somatic mutations occur at a constant rate[32–34]. However, solutions for *DNMT3A*-mutant clones concentrated earlier in life, consistent with early initiation and rapid expansion followed by marked deceleration then slow growth, as

previously mentioned. Of note, capping onset at conception is arbitrary and it remains possible that some clones start later and exhibit faster initial growth followed by even stronger deceleration, a scenario that would be more consistent with published fitness estimates of 11–19% per year based on cross-sectional VAF measurements[13]. By contrast, *SRSF2*[P95H] and *U2AF1* mutations initiated clonal expansion always after 30 years of age and with a median age at onset of 58 and 57 years, respectively (Fig. 4k). This indicates that the reported rarity of these mutant clones[1–3] in people aged less than 60 years is not owing to slow growth over decades, but rather owing to their late onset followed by rapid expansion, and provides a plausible explanation for the high risk of leukaemic progression associated with these mutations[5,40].

## Clonal haematopoiesis dynamics and malignancy

To investigate the links between mutation fitness and malignant progression, we built on our previous study of acute myeloid leukaemia (AML) risk prediction[5], and revealed that among clonal haematopoiesis driver genes, a faster growth rate was associated with a higher AML risk (adjusted $R^2 = 0.55$, $P = 0.0037$; Fig. 5a). For example, genes driving fast clonal haematopoiesis growth—such as *SRSF2* and *U2AF1*—were associated with the highest risks of leukaemogenesis, whereas slow-growing clones—such as those bearing *DNMT3A* mutations—conferred a lower

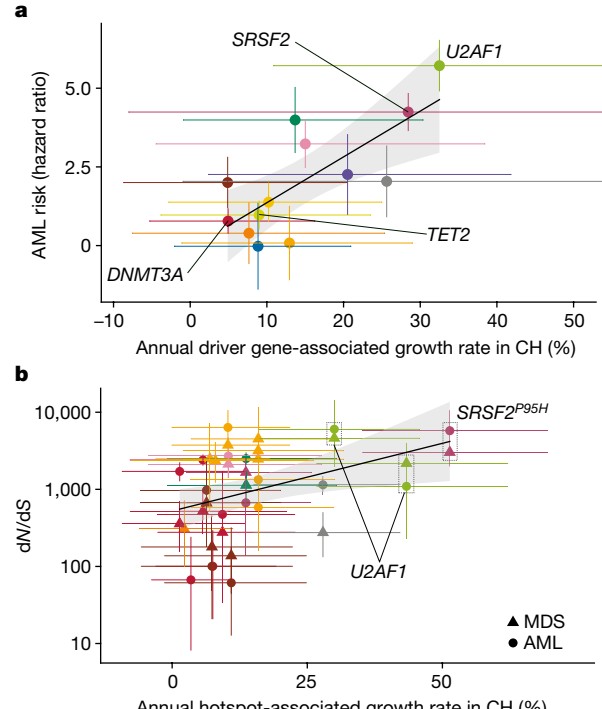

**a**

**b**

**Fig. 5 | Clonal haematopoiesis dynamics and progression to myeloid disease. a**, Relationship between the growth rate associated with each driver gene in clonal haematopoiesis (CH), and the risk of AML progression associated with that driver gene. **b**, Relationship between the growth rate associated with each recurrent mutation in clonal haematopoiesis, and the strength of selection of that mutation in AML (circles) and MDS (triangles). In **a**, **b**, genes and hotspots that are mentioned in the main text are highlighted. AML risk intervals indicate standard error for the estimate[5]; error bars for d$N$/d$S$ show 95% confidence intervals; error bars for annual growth show 90% HPDI. The confidence band (shaded region) represents the 95% confidence interval for the association between annual growth rates and AML risk or d$N$/d$S$.

risk. To confirm our findings in larger studies and include myeloid malignancies other than AML, we analysed large published datasets of AML[41] ($n$ = 1,540) and myelodysplastic syndromes[42] (MDS) ($n$ = 738) using a site-specific extension of the dNdScv algorithm to formally quantify the extent to which individual hotspots are under the influence of positive selection in these cancers[25] (Supplementary Tables 9, 10). This analysis revealed a positive correlation between each hotspot's growth coefficient in clonal haematopoiesis and its selection strength in myeloid cancer (adjusted $R^2$ = 0.19, $P$ = 0.0016; Fig. 5b), corroborating the AML risk analysis. Nevertheless, the observation that the same clonal haematopoiesis driver gene can progress to either AML or MDS, with variable predilections as quantified by gene-level d$N$/d$S$ comparison (Extended Data Fig. 10, Supplementary Table 10), suggests that factors other than growth rate can also influence a mutation's malignant potential.

## Discussion

The phenomenon of clonal haematopoiesis has served as an exemplar in the developing understanding of somatic mutation, clonal selection and oncogenesis in human tissues[10,43]. However, the nature of these interrelated processes can change over time and their consequences develop only slowly, making them difficult to investigate. Here, we studied the longitudinal behaviour of clonal haematopoiesis over long periods (median 13 years) and combined this with lifelong phylodynamic analyses of haematopoiesis to derive new insights into these fundamental biological processes.

First, we found that most clones (92%) display stable exponential growth dynamics in older age, at rates influenced by their driver mutations. This enabled us to predict future clonal growth trajectories, a finding with potentially useful implications for clinical practice (Extended Data Fig. 3f–h). Notably, mutations in *DNMT3A*, reportedly the most common clonal haematopoiesis driver gene[1,2,4], were associated with slower clonal expansion than most other clonal haematopoiesis genes. Also, *DNMT3A* hotspot mutations (for example, at codon R882) were not associated with faster growth than other *DNMT3A* mutations (Fig. 2c). By contrast, *TET2*-mutant clones expanded significantly faster over the study period (Fig. 2c) and, reflecting this, also reached detectable levels much more frequently on-study than *DNMT3A*-mutant clones (Extended Data Fig. 9a). This resulted in *TET2* becoming the most prevalent clonal haematopoiesis driver after the age of 75 years (Fig. 1d).

These findings suggested that, although clonal growth is remarkably stable in old age, dynamics in earlier life may deviate from this behaviour, challenging the premise that mutation fitness is constant over the human lifespan[13]. To test this, we first attempted to derive when individual clonal haematopoiesis clones were founded, using simple retrograde extrapolation of observed trajectories. This led to projected ages at clonal foundation that preceded conception for a large number of clones (Fig. 4f, g), implying that their early growth must have been faster than that we observed during old age. This was most striking for *DNMT3A*, for which more than two thirds of projections were implausible (that is, onset pre-conception), but less common for *TET2* and very uncommon for splicing factor genes (Fig. 4g).

To further investigate lifelong clonal behaviour, we analysed haematopoietic phylogenies from healthy old individuals and found that aged haematopoiesis was dominated by a small number of expanded HSC clones, some of which lacked recognizable drivers[36]. Using phylodynamic approaches to track clonal growth rates through life, in conjunction with findings from our longitudinal cohort, we reveal widespread clonal deceleration prior to the period of stable growth during old age, in the context of an increasingly competitive oligoclonal HSC compartment (Fig. 4i). *DNMT3A*-mutant clones, as well as those bearing mutations in *TP53* and *BRCC3* and also apparently driverless clones, were among those displaying the most marked degree of deceleration (Fig. 4i). The faster growth of *DNMT3A*-mutant clones in early life is supported by comparison with the findings of Watson et al., who analysed cross-sectional VAF spectra from 50,000 individuals and estimated average clonal growth rates across the first 55 years of life; expansion of clones was substantially faster in younger individuals[13] (15.0% per year) compared with older individuals (6.2% per year) (from our study) (Supplementary Table 11). By contrast, *TET2* mutations appeared to drive more stable lifelong growth (Fig. 4h–j), which may underlie their apparent ability to initiate clonal expansion fairly uniformly through life (Fig. 4k) and the fact that *TET2* 'overtakes' *DNMT3A* as the most common clonal haematopoiesis driver after 75 years of age (Fig. 1d and ref. [44]).

In diametric contrast to *DNMT3A* and unlike other genes, clonal haematopoiesis driven by mutant *U2AF1* and *SRSF2*[P95H] initiated only late in life (Fig. 4k) and exhibited some of the fastest expansion dynamics (Fig. 2c). These data were corroborated by phylogenetic analyses (Fig. 3b, f) and tally with the sharp increase in prevalence of splice factor-mutant clonal haematopoiesis[3], MDS[42,45,46] and AML[41,47] in old age and the high risk of progression to myeloid cancers associated with these mutations[5]. The particular behaviour of these clones suggests a specific interaction with ageing, which could relate to cell-intrinsic factors or to cell-extrinsic changes in the aging haematopoietic niche that favour splice factor mutations[48,49].

Finally, we explored the relationship between clonal growth rate in clonal haematopoiesis and the development of myeloid cancers. We find that mutations associated with faster clonal haematopoiesis growth are also those associated with higher risk of progression to AML (Fig. 5a) and are under the strongest selective pressure in AML

and MDS (Fig. 5b). Indeed, we show that the average annual growth per gene explains more than 50% of the variance in AML risk progression. This shows that an improved understanding of growth dynamics in clonal haematopoiesis can help identify those at risk of myeloid malignancies.

Collectively, our work gives new insights into the lifelong clonal dynamics of different subtypes of clonal haematopoiesis, the impact of ageing on haematopoiesis, and the processes linking somatic mutation, clonal expansion and malignant progression.

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

# Methods

## Study participants

Ethical permission for this study was granted by The East of England (Essex) Research Ethics Committee (REC reference 15/EE/0327). The SardiNIA longitudinal study recruited individuals from four towns in the Lanusei Valley in Sardinia, capturing 5 phases of sample and data collection[26] over more than 20 years. Informed consent was obtained from all participants. We analysed serial samples from 385 individuals in the SardiNIA project.

## Targeted sequencing and variant calling

Target enrichment of whole-blood DNA was performed using a custom RNA bait set (Agilent SureSelect ELID 3156971), designed complementary to 56 genes implicated in clonal haematopoiesis and haematological malignancies (Supplementary Table 1). Libraries were sequenced on Illumina HiSeq 2000 and variant calling was performed as we described previously[5,50]. In brief, somatic single-nucleotide variants and small indels were called using Shearwater (v.1.21.5), an algorithm designed to detect subclonal mutations in deep sequencing experiments[51]. Two additional variant-calling algorithms were applied to complement this approach: CaVEMan (v.1.11.2) for single-nucleotide variants, and Pindel (v.2.2) for small indels[52,53]. VAF correction was performed using an in-house script (https://github.com/cancerit/vafCorrect). Finally, allele counts at recurrent mutation hotspots were verified using an in-house script (https://github.com/cancerit/allelecount). Variants were filtered as we described previously[5,50], but were not curated with regard to existing notions of oncogenicity, that is, all somatic variants passing quality filters were retained for analysis.

If a variant was identified in an individual at any time point in the study, this site was re-queried in the same individual at all other time points, using an in-house script (cgpVAF) to provide pileup (SNV) and Exonerate (indel) output (https://github.com/cancerit/vafCorrect). No additional filters were applied to these back-called variants.

## Selection analyses

To quantify selection, we used the dNdScv algorithm, a maximum-likelihood implementation of d$N$/d$S$, which measures the ratio of non-synonymous ($N$) to synonymous ($S$) mutations, while controlling for gene sequence composition and variable substitution rates[27]. We first applied this method to the mutation calls from the longitudinal SardiNIA cohort in order to identify which genes are under positive selection in the context of clonal haematopoiesis. For this analysis, any mutation that was present in a single individual at multiple time points was counted only once. We also compared d$N$/d$S$ ratios at the beginning and end of study, and found the latter to be higher, consistent with stronger cumulative effects of selection at older ages (Supplementary Note 2).

To characterize patterns of selection in AML and MDS, we applied dNdScv to two published data sets. The AML set was derived from 1,540 patients enrolled in three prospective trials of intensive therapy[41]. The MDS set included 738 patients with MDS or closely related neoplasms such as chronic myelomonocytic leukaemia[42]. Both used deep targeted sequencing of 111 cancer genes, which overlapped with 13 of the 17 genes of interest in our longitudinal clonal haematopoiesis study (*PPM1D*, *CTCF*, *GNB1* and *BRCC3* were not sequenced in the AML or MDS studies). We called and filtered variants in the 13 overlapping genes using the strategy described above ('Targeted sequencing and variant calling'). Variants were identified in all 13 genes in both AML and MDS datasets (Supplementary Table 10). We calculated d$N$/d$S$ values both at the level of individual genes, and at single-site level for hotspots, the latter using the sitednds function in the dNdScv R package.

Finally, we compared d$N$/d$S$ ratios in shared and private branches of the three phylogenies, and found selection to be stronger in the former, consistent with the fact that mutations along shared branches were the ones driving subsequent clonal expansions (and therefore were more strongly selected) (Supplementary Note 2).

## Modelling of clone trajectories through time

We use Bayesian hierarchical modelling to model clonal trajectories. Since we are unable to reliably phase different mutations into specific clones (Supplementary Note 3) and given that individual clonal haematopoiesis clones typically harbour a single driver mutation[54], we assume that each mutation is heterozygous and its VAF is representative of the prevalence of a single clone. Accordingly, for a given individual $j$ and mutation $i$, we have a mutant clone $c_{ij}$. We model the counts $\text{counts}_{c_{ij}}$ for $c_{ij}$ at age $t$ as a binomial distribution (Bin), such that $\text{counts}_{c_{ij}}(t) \sim \text{Bin}(\text{cov}_{ij}(t), p_{ij}(t))$, with $\text{cov}_{ij}$ as the coverage of this mutation at age $t$ and $p_{ij}(t) \sim \text{Beta}(\alpha(t), \beta)$ as the expected proportion of mutant allele copies. As such, $\text{counts}_{c_{ij}}(t) \sim \text{BB}(\text{cov}_{ij}(t), \alpha(t), \beta)$, where BB is the beta binomial distribution. Here, $\beta \sim N(\mu_{od}, \sigma_{od})$ is the technical overdispersion parameterized as a normal distribution whose parameters ($\mu_{od}$ and $\sigma_{od}$, the mean and standard deviation, respectively) are estimated using replicate data (details below) and $\alpha(t) = \frac{\beta q(t)}{1 - q(t)}$, where $q(t) = \text{ilogit}((b_{\text{gene}_i} + b_{\text{site}_i} + b_{c_{ij}}) \times t + u_{ij})$, with ilogit representing the inverse function. We use this parameterization to guarantee that $E[\text{counts}_{c_{ij}}] = p_{ij}\text{cov}_{ij}$. $b_{\text{gene}_i} \sim N(0,0.1)$ and $b_{\text{site}_i} \sim N(0,0.1)$ are the gene and site growth effects for mutation $i$, respectively. $b_{c_{ij}} \sim N(0,0.05)$ is the growth effect associated exclusively with mutation $i$ in individual $j$—that is, of mutant clone $c_{ij}$—and $u_{ij}$ is the offset accounting for the onset of different clones at different points in time. We also define the growth effect of $c_{ij}$ as $b_{\text{total}_{ij}} = (b_{\text{gene}_i} + b_{\text{site}_i} + b_{uc_{ij}})$. Throughout this work we will refer to $b_{\text{gene}_i} + b_{\text{site}_i}$ as the driver (growth) effect and to $b_{c_{ij}}$ as the unknown-cause (growth) effect—the fraction of growth that is quantifiable but not explained by the driver mutation, and is attributable to other factors that may affect clonal growth, but differ between individuals, such as age, sex, interclonal competition and others.

**Preventing identifiability issues and reducing uninformed estimates.** To address possible identifiability issues in our model, when a gene has a single mutation (*JAK2*^V617F^ and *IDH2*^R140Q^), the effect is considered to occur only at the site level. To avoid estimating the dynamics of a site from a single individual, we only model $b_{\text{site}_i}$ when two or more individuals have a missense mutation on site $i$, we refer to these sites as 'recurrent sites'. Overall, we consider a total of 17 genes and 39 recurrent sites (Supplementary Table 5).

**Estimating and validating growth parameters.** Using the model described above, we use Markov chain Monte Carlo (MCMC) with a Hamiltonian Monte Carlo (HMC) sampler with 150–300 leapfrog steps as implemented in greta[55]. We sample for 5,000 iterations and discard the initial 2,500 to get estimates for the distribution of our parameters. As such, our estimates for each parameter are obtained considering their mean, median and 95% highest density posterior interval for 2,500 samples.

We assess the goodness-of-fit using the number of outliers detected in any trajectory and consider only trajectories with no outliers as being explained by our model and, as such, growing at constant rate. Outliers are assessed by calculating the tail probabilities of the counts under our model with a hard cut-off at 2.5%. Thus, $P_{\text{outlier}} = 1$ if $P(\text{counts} \mid b_{\text{gene}_i}, b_{\text{site}_i}, b_{c_{ij}}, u_{ij}, t) < 0.025 \mid P(\text{counts} \mid b_{\text{gene}_i}, b_{\text{site}_i}, b_{c_{ij}}, u_{ij}, t) > 0.975$ and $P_{\text{outlier}} = 0$ otherwise. We validate this approach using Wright–Fisher simulations (Supplementary Methods). We additionally assess the predictive power of this model on an additional time point that was available for a subset of individuals and that was not used in the inference of parameters in our model (Supplementary Methods).

**Estimating the technical overdispersion parameter.** Technical VAF overdispersion used two distinct sets of data:

(1) Horizon Tru-Q-1 was serially diluted to VAFs of 0.05, 0.02, 0.01,

0.005 and 0 using Horizon Tru-Q-0 (verified wild-type at these variant sites), then sequenced in duplicate or triplicate;
(2) 19 SardiNIA samples with mutations across 15 genes at a range of VAFs, were sequenced in triplicate.

Sample processing and analysis was performed as described in 'Targeted sequencing and variant calling' section. Replicate samples were picked from the same stock of DNA, then library preparation and sequencing steps were performed in parallel. Variant calls for these replicate samples are in Supplementary Table 12.

For (1), we model the distribution over the expected *VAF* as a beta distribution such that $VAF \sim \text{Beta}(\alpha, \beta)$ and for (2) we adopt a model identical to the one described earlier in this section but use only gene growth effects ($\text{counts}_{c_{ij}}(t) \sim \text{BB}(\text{cov}_{ij}(t), \alpha(t), \beta)$, $\alpha(t) = \frac{\beta q(t)}{1 - q(t)}$, $q(t) = \text{ilogit}(b_{\text{gene}_i} \times t + u_{ij})$). Here, we model $\beta \sim \exp(r)$ with $r$ as a variable with no prior. We use MCMC with HMC sampling with 400–500 leapfrog steps as implemented in greta[55] to estimate the mean and standard deviation of $\beta$. For this estimate we use 1,000 samples from the posterior distribution.

## Non-mutation factors and clonal growth rate

**Inherited polymorphisms and JAK2-mutant clonal growth.** The SardiNIA cohort had previously been characterized using two Illumina custom arrays: the Cardio-MetaboChip and the Immuno-Chip[26]. Inherited genotypes at 12 loci previously associated with MPN risk were extracted for the 12 individuals with *JAK2[V617F]* mutation[23,24]. The relationship between each individual's total number of inherited risk alleles and *JAK2*-mutant clonal growth rate was assessed by Pearson's correlation. The 46/1 haplotype, which harbours 4 SNPs in complete linkage disequilibrium, was considered as a single risk allele.

**Age, sex and smoking experience.** We assess the association between unknown-cause growth and age through the calculation of a Pearson correlation considering all genes, both together and separately while controlling for multiple testing. We also assess the association between unknown-cause growth and sex and smoking history using a multivariate regression where unknown-cause growth is the dependent variable and sex and previous smoking experience are the covariates, while also controlling for age.

## Determining the age at clone onset

We consider that HSC clones grow according to a Wright–Fisher model. According to this, for an initial population of HSC $n/2$, we can consider two scenarios—that of a single growth process where the time at which the cell first starts growing $t_0$ is described as $t_0 = \frac{\log\left(\frac{1}{n}\right) - u}{b_{\text{total}}}$, or that of a two-step growth process, where $t_0\text{adjusted} = t_0 + \frac{\log(g/b_{\text{total}})}{b_{\text{total}}} - \frac{1}{b_{\text{total}}}$, where $g$ is the number of generations per year. The latter scenario is the one chosen, due to its strong theoretical foundation and previous application to mathematical modelling of cancer evolution[56]. The two regimes that describe it are an initial stochastic growth regime and, once the clone reaches a sufficient population size, a deterministic growth regime. The adjustment made to $t_0$ in $t_0\text{adjusted}$ can be interpreted as first estimating the age at which the clone reached the deterministic growth phase ($t_0 + \frac{\log(g/b_{\text{total}})}{b_{\text{total}}}$) followed by subtracting the expected time for a clone to overcome its stochastic growth phase ($\frac{1}{b_{\text{total}}}$). For both $n$ and $g$ we use the estimates based on ref. [33]: $n = 50,000$ and $g = 2$. We validate this approach using simulations (Supplementary Methods) and test the approach against our serial VAF data and verify that changes in $n$ and $g$ do not have a marked effect on age at onset estimates by considering a range of values ($n = \{10,000; 50,000; 100,000; 200,000; 600,000\}$ and $g = \{1; 2; 5; 10; 13; 20\}$).

## Cell colonies and phylogenetic trees

**Sample preparation and sequencing.** We selected 3 individuals with splicing gene mutations from the SardiNIA cohort for detailed blood phylogenetic analysis. Peripheral blood samples were drawn into Lithium-heparin tubes (vacutest, kima, 9 ml) and buccal samples were taken (Orangene DNA OG-250). Peripheral blood mononuclear cells were isolated from blood and plated at 50,000 cells per ml in MethoCult 4034 (Stemcell Technologies). After 14 days in culture, 96 single haematopoietic colonies were plucked per individual (total 288 colonies, each made up of hundreds to thousands of cells) and lysed in 50 μl of RLT lysis buffer (Qiagen).

Library preparation for WGS was performed using our low-input pipeline as previously described[57,58]. The 150 bp paired-end sequencing reads were generated using the NovaSeq 6000 platform to a mean sequencing depth of 15× per sample. Reads were aligned to the human reference genome (NCBI build37) using BWA-MEM.

**Variant calling and filtering.** Single-nucleotide variants (SNVs) and small indels were called against an unmatched reference genome using the in-house pipelines CaVEMan and Pindel, respectively[52,53]. 'Normal contamination of tumour' was set to 0.05; otherwise, standard settings and filters were applied. For all mutations passing quality filters in at least one sample, in-house software (cgpVAF, https://github.com/cancerit/vafCorrect) was used to produce matrices of variant and normal reads at each mutant site for all colonies from that individual. Copy-number aberrations and structural variants were identified using matched-normal ASCAT[59] and BRASS (https://github.com/cancerit/BRASS). Low-coverage samples (mean <4×) were excluded from downstream analysis ($n = 1$, PD41305). Samples in which the peak density of somatic mutation VAFs was lower than expected for heterozygous changes (in practice VAF < 0.4) were suspected to be contaminated or mixed colonies, and were also excluded from further analysis ($n = 3$, PD41305; $n = 9$, PD41276; $n = 3$, PD34493).

Multiple post-hoc filtering steps were then applied to remove germline mutations, recurrent library prep or sequencing artefacts, and in vitro mutations, as described previously[60] and detailed in custom R scripts (https://github.com/margaretefabre/Clonal_dynamics). Buccal samples were used as an additional filter; mutations were removed if the variant:normal count in the buccal sample was consistent with that expected for a germline mutation (0.5 for autosomes and 0.95 for X and Y chromosomes, binomial probability >0.01), and were retained if (1) the variant:normal count in the buccal sample was not consistent with germline (binomial probability $<1 \times 10^{-4}$) and (2) the mutation was not present in either of 2 large SNP databases (1000 Genomes Project and Kaviar) with MAF > 0.001.

**Phylogenetic tree construction and assignment of mutations back to the tree.** These steps were also performed as described previously[60] and are detailed here: https://github.com/margaretefabre/Clonal_dynamics. In brief, samples were assigned a genotype for each mutation site passing filtering steps ('present' = ≥2 variant reads and probability > 0.05 that counts came from a somatic distribution; 'absent' = 0 variant reads and depth ≥6; 'unknown' = neither 'absent' nor 'present' criteria met). The proportion of 'unknown' genotypes going into tree-building was low: 1.5% (PD34493), 1.4% (PD41276) and 1.3% (PD41305; Extended Data Fig. 5a–c). A genotype matrix of shared mutations was fed into the MPBoot program[61], which constructs a maximum parsimony phylogenetic tree with bootstrap approximation. The in-house-developed R package treemut (https://github.com/NickWilliamsSanger/treemut), which uses original count data and a maximum likelihood approach, was then used to assign mutations back to individual branches on the tree. Since individual edge length is influenced by the sensitivity of variant calling, lengths were scaled by 1/sensitivity, where sensitivity was calculated as the proportion of germline variants called (mean sensitivity: 85.4%, 87.0% and 83.5% for PD41305, PD41276 and PD34493, respectively). The approaches we used to validate the phylogenies, including comparison of MPBoot with an alternative phylogeny-inference algorithm, SCITE[62], are detailed in Supplementary Methods.

**Reconstruction of population trajectories.** Phylogenies were made ultrametric (branch lengths normalized) using a bespoke R function (make.tree.ultrametric, https://github.com/margaretefabre/Clonal_dynamics/my_functions). With the root of the tree representing conception and the tips representing age at sampling, we scaled the age axis in two phases by: (1) assigning the first 55 mutations to the period between conception and birth (in light of evidence for this higher rate of mutation acquisition during this period[36,60], and (2) scaling the axis linearly throughout life after birth (in light of evidence for a constant rate of mutation acquisition in HSCs during postnatal life[32–37]. We then analysed population size trajectories by fitting Bayesian nonparametric phylodynamic reconstructions (BNPR) as implemented in the phylodyn R package[38,39] to clades - sets of samples in a phylogenetic tree sharing a most recent common ancestor (MRCA)—defined by either having a driver mutation on the MRCA or a MRCA branch length that spans more than 10% of the tree depth and with 5 tips or more. We also estimated the lower and upper bounds for age at onset of clonal expansion to be the limits of the branch containing the most recent common ancestor.

### Detection of clonal deceleration
We detect deceleration using two different approaches—the ratio between expected and observed clone size using phylodynamic estimates and the ratios between observed and historical (from longitudinal data) and between late and expected (from phylogenetic data), respectively. To obtain the late growth rate we fit a biphasic log-linear model to our phylodynamic estimation of $N_{\text{eff}}$—this enables us to obtain an early and a late growth rate (details in the Supplementary Methods).

**Expected and observed clone size.** The expected clone size is calculated by extrapolating the early growth rate until the age of sampling; having this we can calculate the ratio between expected and observed growth. The ratio between these quantities is then used as a measure of deceleration (details in the Supplementary Methods).

**Growth ratio in phylogenetic data.** The late growth rate is defined as the late growth rate defined in the previous section of the methods. The expected growth rate for the phylogenies is calculated as the growth coefficient for a sigmoidal regression that assumes a population size of 200,000 HSC as the carrying capacity. We then use the ratio between these quantities as a measure of deceleration (1 implies no deceleration; <1 implies deceleration).

**Growth ratio in longitudinal data.** The observed growth rate is defined as the growth rate inferred directly from the data. The minimal historical growth is the growth rate estimate obtained by restricting clone initiation to a time after conception (age at at onset > −1).

### Clonal haematopoiesis dynamics and malignant progression
To calculate the association between clonal haematopoiesis dynamics and AML we used the risk coefficients from our previous work in predicting the onset of AML[5], which were calculated by fitting a Cox-proportional hazards model that calculated the risk of AML onset associated with each gene (agnostic of clone size) while controlling for age, sex and cohort, and estimate the coefficient of correlation between the expected value of the annual growth for the posterior distribution of each gene (considering gene, site and unknown-cause effects) and the AML progression risk.

The association between clonal haematopoiesis dynamics and selection in MDS and AML use the d$N$/d$S$ values calculated with dNdScv as previously described in the methods, using two distinct cohorts from previous studies[41,42]. d$N$/d$S$ values were calculated for all hotspots and their coefficient of correlation with the expected value of the annual growth for the posterior distribution of each hotspot (also considering gene, site and unknown-cause effects) was calculated.

### Statistical analyses
All statistical analyses were conducted using the R software[63] - MCMC models were fitted using greta[55] and hypothesis testing, generalized linear models and maximum likelihood fits were performed in base R.

### Reporting summary
Further information on research design is available in the Nature Research Reporting Summary linked to this paper.

### Data availability
The data files necessary to run the analysis in https://github.com/josegcpa/clonal_dynamics are freely available at https://doi.org/10.6084/m9.figshare.15029118. All sequencing data have been deposited in the European Genome–phenome Archive (EGA) (https://www.ebi.ac.uk/ega/). Targeted sequencing data have been deposited with EGA accession numbers EGAD00001007682 and EGAD00001007683; WGS data have been deposited with accession number EGAD00001007684. Data from the EGA are accessible for research use only to all bona fide researchers, as assessed by the Data Access Committee (https://www.ebi.ac.uk/ega/about/access). Data can be accessed by registering for an EGA account and contacting the Data Access Committee.

### Code availability
All analyses reported in this study used the statistical software R (v.3.6.3). All R files used for the longitudinal and phylodynamic modelling and validation are publicly available at https://github.com/josegcpa/clonal_dynamics. All files used for the construction of phylogenetic trees are publicly available at https://github.com/margaretefabre/Clonal_dynamics.

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

**Acknowledgements** This work was funded by a joint grant from the Leukemia and Lymphoma Society (RTF6006–19) and the Rising Tide Foundation for Clinical Cancer Research (CCR-18–500) and by the Wellcome Trust (WT098051). M.A.F. is funded by a Wellcome Clinical Research Fellowship (WT098051). J.G.d.A. is supported by the NIHR Cambridge BRC and their opinions are not necessarily those of the NHS, the NIHR or the Department of Health and Social Care. G.S.V. is funded by a Cancer Research UK Senior Cancer Fellowship (C22324/A23015) and work in his lab is also funded by the European Research Council, Kay Kendall Leukaemia Fund, Blood Cancer UK and the Wellcome Trust. E.F.M. is supported by the Wellcome Trust and Beit Foundation (104064/Z/14/Z) and by the EC H2020. The collection of samples and data from the SardiNIA longitudinal cohort study was supported by the Intramural Research Program of the NIH, National Institute on Aging (NIA) of the National Institute of Health (NIH) with contracts N01-AG-1- 2109 and HHSN271201100005C; and by the

European Union's Horizon 2020 Research and Innovation Programme under grant agreement 633964 (ImmunoAgeing). We thank J. Blundell and C. Watson for helpful discussions relating to comparison between the findings in this manuscript and their published work[13].

**Author contributions** G.S.V. and M.G. conceived and supervised the study. M.A.F., J.G.d.A. and M.S.V. carried out analyses and generated data figures. M.G. and J.G.d.A. developed and implemented the statistical modelling of clonal dynamics. V.O., E.F., M.M. and F.C. oversaw the SardiNIA cohort. V.O., E.F., M.M., E.F.M. and F.C. provided samples and data from the Immunoageing study. A.D., J.R., C.H., J.B., M.A.F. and G.S.V. processed participant samples and performed assays. F.A., N.W., J.N. and I.M. generated computational code used in this paper. E.M., M.S.C. and P.J.C. provided single-cell-derived colony WGS data and helped with data analysis/interpretation.

**Competing interests** G.S.V. is a consultant for STRM.BIO and receives a research grant from Astrazeneca. The other authors declare no competing interests.

**Additional information**
**Correspondence and requests for materials** should be addressed to Moritz Gerstung or George S. Vassiliou.

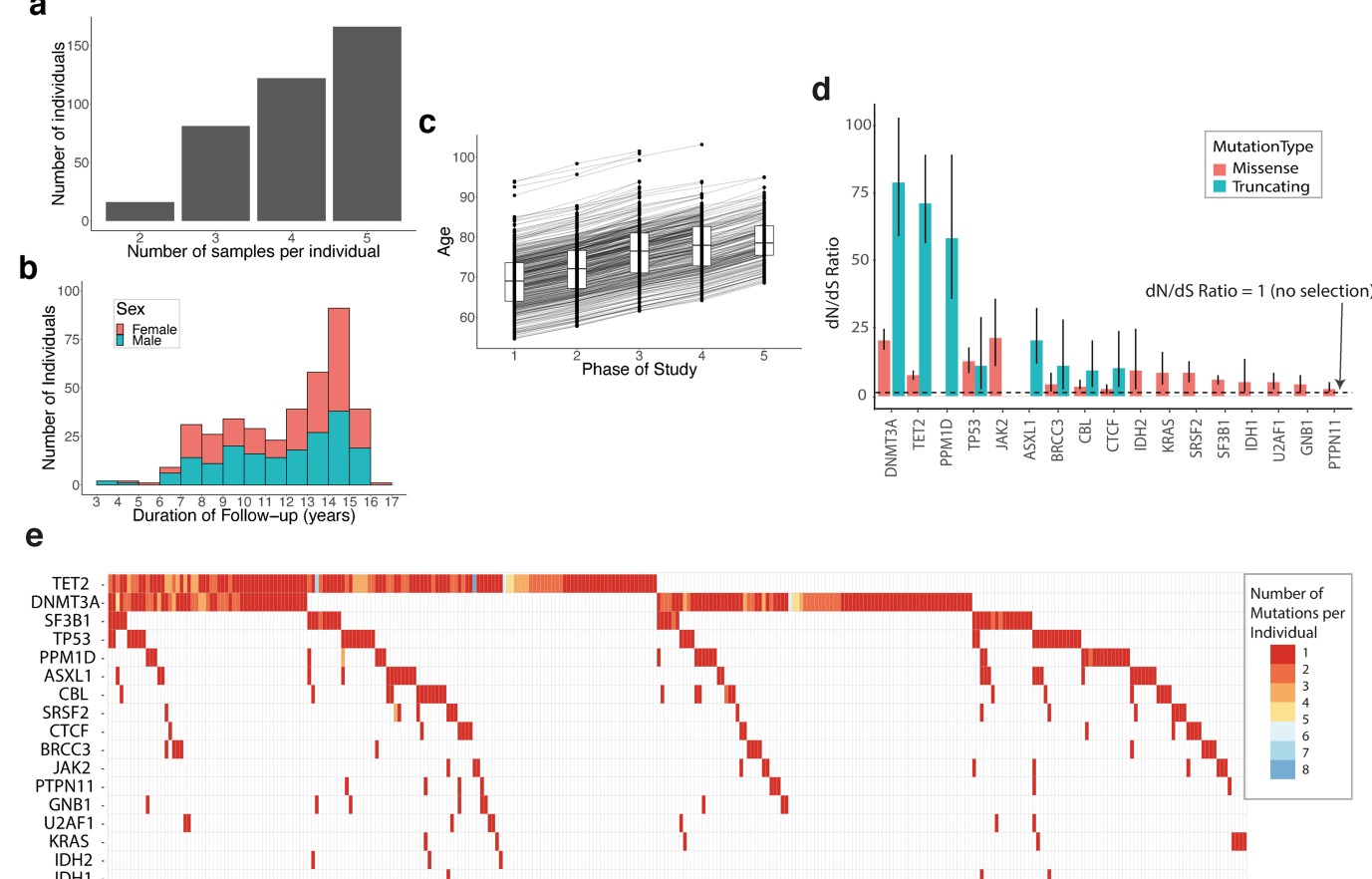

**Extended Data Fig. 1 | Longitudinal cohort characteristics and mutation prevalence and selection across the studied genes. a**, Distribution of the number of serial samples obtained per individual. **b**, Duration of follow-up per individual. **c**, Distribution of participants' ages at each of the five sampling phases of the SardiNIA study. The boxes represent the 25th, 50th (median) and 75th percentiles of the data; the whiskers represent the lowest (or highest) datum within 1 interquartile range from the 25th (or 75th) percentile. **d**, Observed-to-expected (dN/dS) ratios for the 17 genes with missense and/or truncating mutations under positive selection (with q < 0.1). The dashed line

indicates a dN/dS value of 1, which represents neutrality (no selection). Error bars depict 95% CIs. **e**, Waterfall plot showing the number and distribution of mutations among participants. Each column represents 1 individual, and each row 1 gene. Coloured squares indicate the presence of a mutation with the specific colour indicating the number of distinct mutations in that gene identified in that individual. For individuals with the same mutation identified at multiple serial time-points, the serially-observed mutation is counted only once.

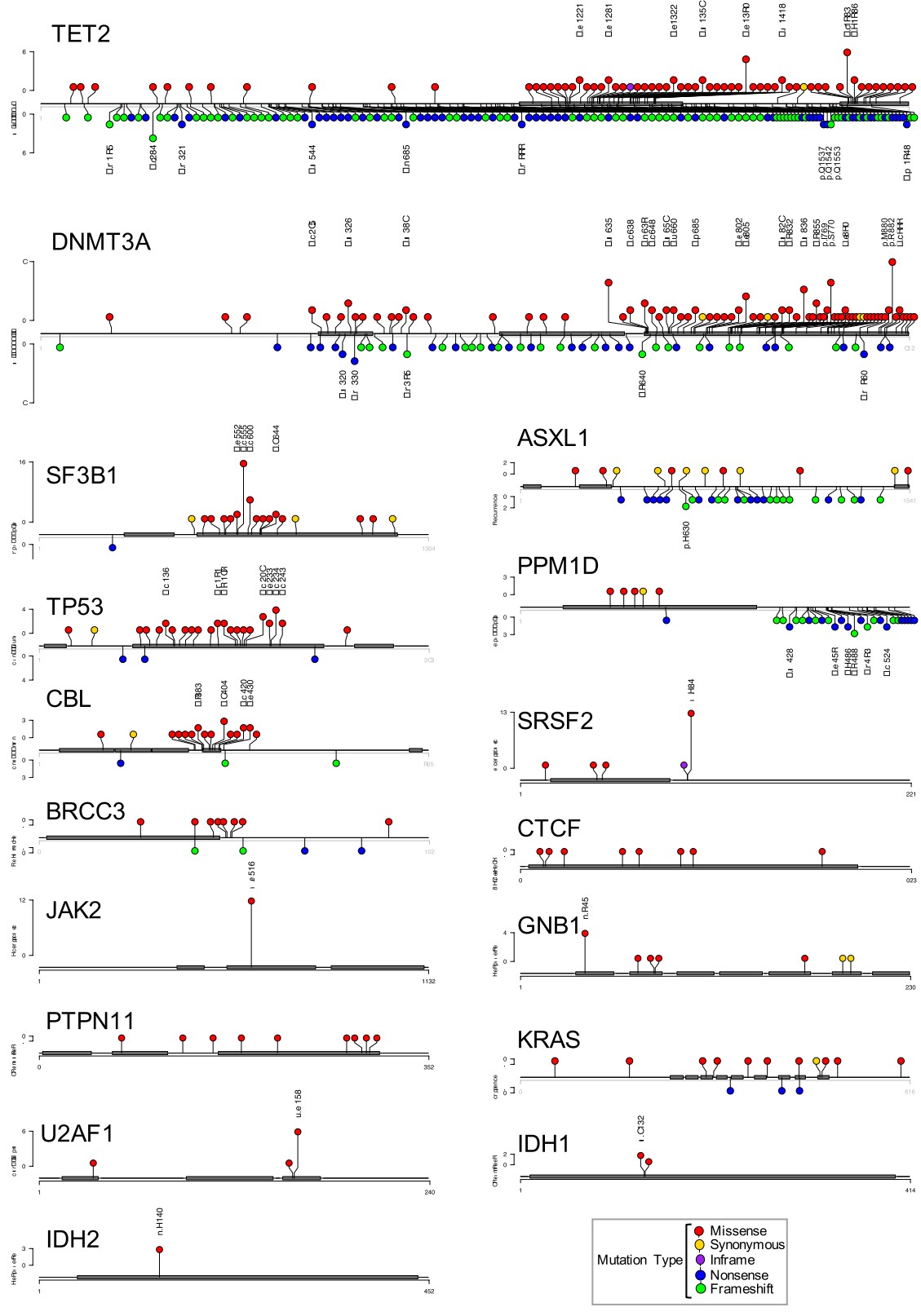

**Extended Data Fig. 2 | Distribution of somatic mutations within driver genes (previous page).** Lolliplots show the longest protein isoform of each gene, with protein domains depicted by grey rectangles. Each circle represents a somatic mutation. The vertical distance of the circle from the protein cartoon indicates its recurrence in the cohort (quantified on the y-axis). Amino acid codons recurrently mutated (ie. observed in more than one individual) in our cohort are explicitly labelled. Circle colours indicate the mutation type as per key. Non-truncating mutations (missense, inframe, synonymous) are depicted above and truncating mutations (nonsense, frameshift) below the protein cartoon.

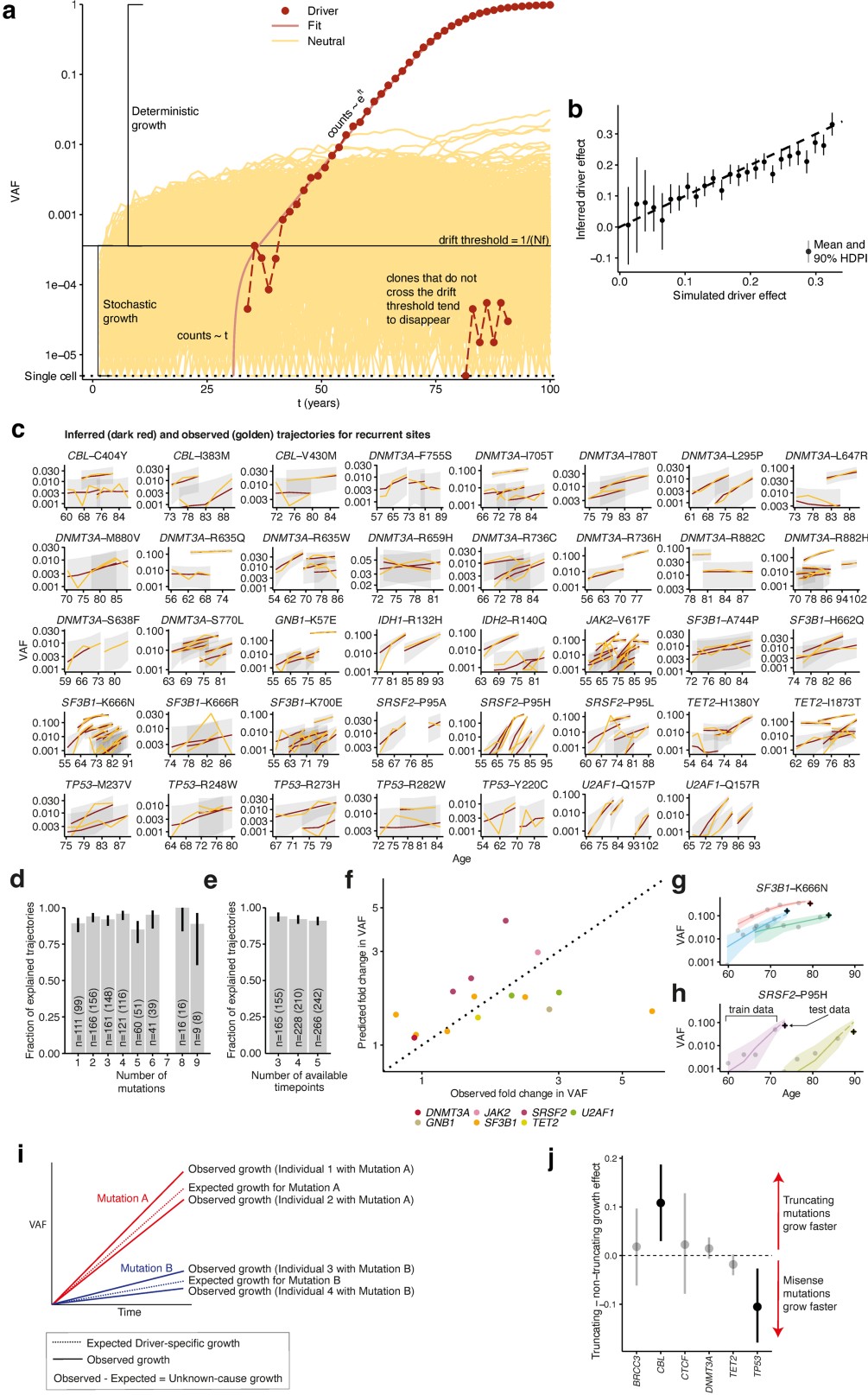

**Extended Data Fig. 3** | See next page for caption.

**Extended Data Fig. 3 | Modelling CH dynamics in older age using time-series VAF data (previous page). a**, Representation of a Wright-Fisher simulation, showing two phases of clonal growth. The likelihood of a clone transitioning from stochastic to deterministic growth is inversely proportional to the product of its fitness (f) and the total number of stem cells (N). Clones with no fitness advantage (depicted in yellow) are unlikely to exceed their drift thresholds and tend to disappear or remain undetectable. Fitter clones (depicted in red) are more likely to reach deterministic growth. **b**, Association between the driver mutation effect used in the Wright-Fisher simulations and the driver effect inferred using our model ($R^2 = 0.92$; n = 270 simulated clones). Error bars represent 90% highest posterior density interval (HDPI). **c**, Comparison of observed (golden) and inferred (mean estimate; red) trajectories for all recurrently mutated sites. Grey bands represent 95% highest posterior density intervals. **d**, Relationship between the number of mutations co-occurring within an individual and the proportion of clones growing at a fixed rate over time (n = 685 clones; the number of clones used to calculate each ratio estimate is represented on each bar and in brackets is the number of explained trajectories). **e**, Relationship between the number of available timepoints in a trajectory and the proportion of clones growing at a fixed rate over time (n = 659 clones; the number of clones used to calculate each ratio estimate is represented on each bar and in brackets is the number of explained trajectories). Error bars represent the beta-distributed 90% confidence intervals (in **d** and **e**). **f**, Association between predicted and observed VAF in additional prospectively-collected samples from 11 individuals with 15 CH driver mutations, not used for growth rate inference. The dotted line depicts theoretical perfect agreement between predicted and observed VAF. **g**,**h**, Example trajectories of clones with *SF3B1*-K666N (f) and *SRSF2*-P95H (g) mutations. Points represent VAFs used in our model to fit the growth curve (train), and crosses represent prospectively tested VAFs used (test), showing good agreement between predicted and observed VAFs. Bands represent the 95% HPDI. **i**, Illustration of the determinants of growth in our model. Each mutation drives an expected rate of clonal growth. **j**, Comparison of growth rate associated with truncating vs non-truncating mutations in genes with both driver types. Points above the dashed line show faster growth for truncating mutations, and points below show faster growth for non-truncating mutations (n = 514 clones). Intervals represent the 90% HPDI for the difference between truncating and non-truncating mutations.

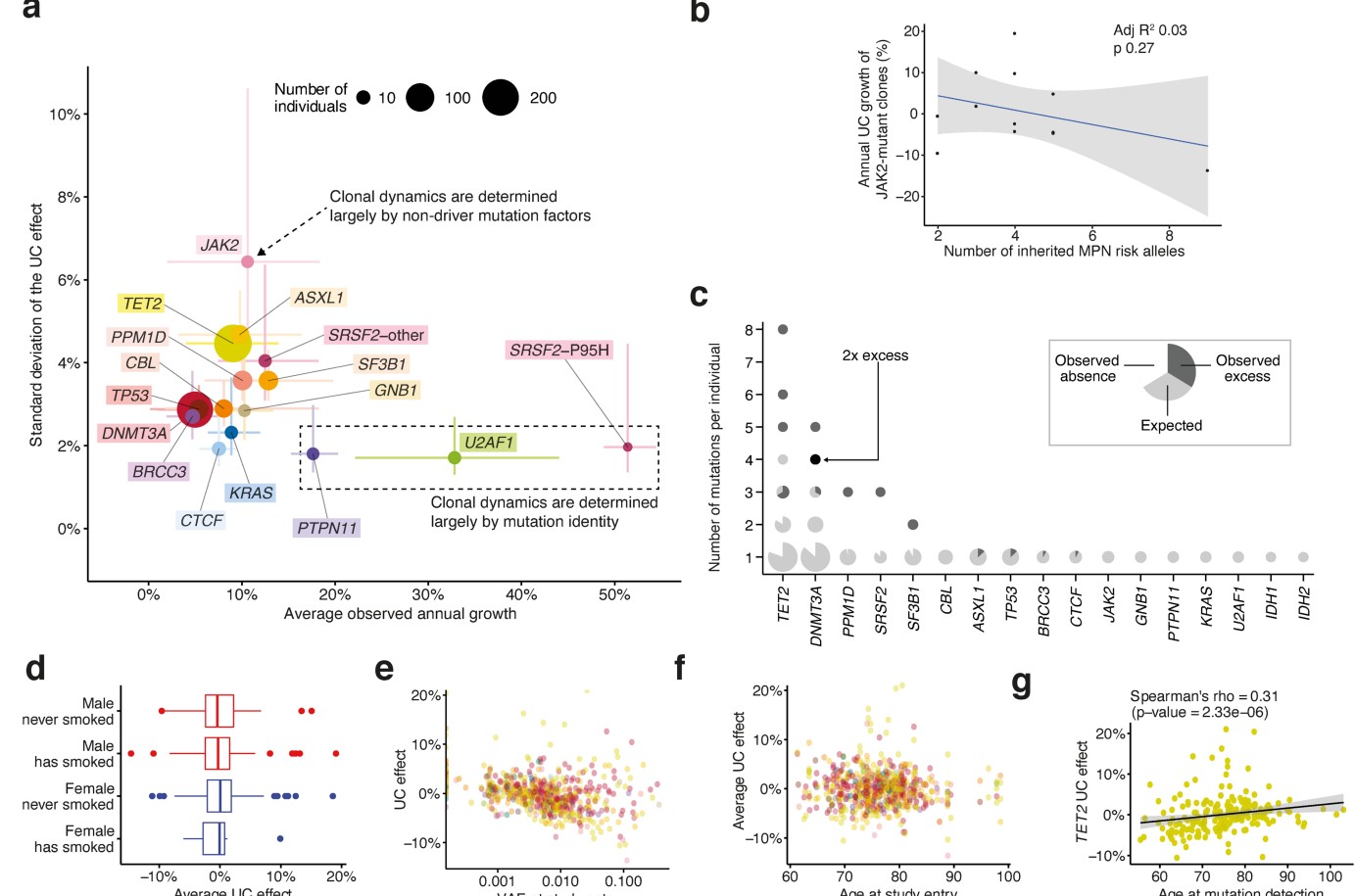

**Extended Data Fig. 4 | Differences in growth rate between individuals/clones with the same driver. a**, For each gene, we contrast the mean annual growth rate among individuals/clones bearing a mutation in that gene, with the spread in this rate (defined here as the standard deviation of the unknown-cause (UC) growth). Circles represent point estimates, with circle size indicating the number of clones bearing a mutation in that gene, and lines representing the 90% confidence interval (CI). For the standard deviation, the 90% CI was calculated assuming that $\frac{(n-1)s^2}{\sigma^2} \sim Chisq(n-1)$, with $n$ being the sample size, $s$ the standard deviation estimate and $\sigma^2$ the true population variance. *SRSF2*-P95H mutations are plotted separately to other SRSF2 mutations, as they are associated with significantly different growth dynamics (n = 633 clones). **b**, Relationship between number of inherited MPN risk alleles and *JAK2*-mutant clonal growth rate (Pearson $R^2$ = 0.03; p = 0.27 (two-sided)). The grey band represents the 95% confidence interval for the linear regression. **c**, The number of mutations per individual in each gene is plotted. Each data-point is a pie-chart, the size of which reflects the number of individuals. For each gene, given the observed mutation prevalence in our cohort, the pie is fully light grey if the number of individuals we observed with the specific number of mutations is the same as the number of individuals we expected by

chance. The presence of a white segment indicates that we found fewer individuals with that number of mutations than expected. The presence of a dark grey segment indicates that we found an excess of individuals with that number of mutations. We estimate the expected number of mutations in each gene in each individual through Monte Carlo estimation; assuming the prevalence of mutations in the cohort is uniform for each gene across individuals, we simulate 1,000 scenarios where we randomly distribute these mutations given the number of mutations in each individual. **d**, Association between sex and smoking history and the average UC effect for each individual (n.s.; n = 628 clones). The boxes represent the 25th, 50th (median) and 75th percentiles of the data; the whiskers represent the lowest (or highest) datum within 1 interquartile range from the 25th (or 75th) percentile. **e**, Association between VAF at study entry and the average UC effect for each individual ($R^2$ = 0.062; $CI_{95\%}$ = [0.029,0.107]; p = 2.42*$10^{-9}$). **f**, Association between age at study entry and the average UC effect for each individual (n.s.). **g**, Association between age at mutation detection and UC effect for each *TET2*-mutant clone (Spearman's rho = 0.31; p = 2.33*$10^{-6}$ (two-sided)). The grey band represents the 95% confidence interval for the linear regression.

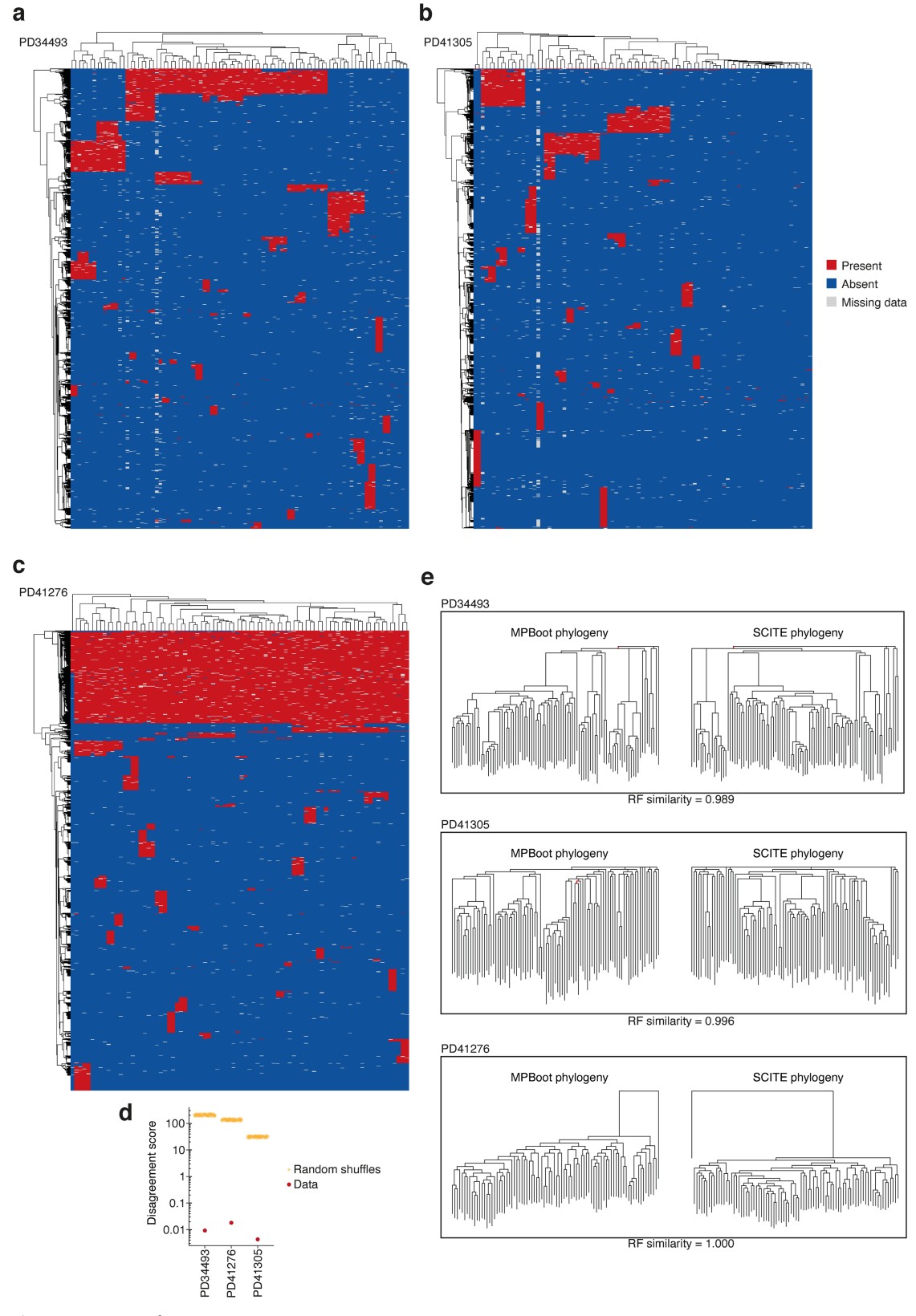

**Extended Data Fig. 5 |** See next page for caption.

**Extended Data Fig. 5 | Data quality and validation of phylogenetic trees.**
**a**–**c**, Heatmaps of the genotype data used for tree inference for the three individuals for which trees were derived in our study (PD34493, PD41305 and PD41276, respectively), with colours corresponding to the presence (red), absence (blue) and uncertainty (grey) of each genotype (rows) across all colonies (columns). For both colonies and genotypes, dendrograms derived from the hierarchical clustering of each are shown and are not representative of the derived phylogenetic trees. **d**, Internal consistency of the shared mutation data for each individual as determined by the disagreement score. A perfect phylogeny has a score of zero. We compare scores for the data with scores for random shuffles of the genotype data at each locus. **e**, Comparison of phylogenetic trees built by alternative phylogeny-inference algorithms, MPBoot and SCITE, for each of the 3 individuals. For all three we present the Robinson-Fould (RF) similarity between trees built by the two methods, with 0 representing completely different trees and 1 representing identical trees. Branching events that are different between trees constructed using the two methods are highlighted in red.

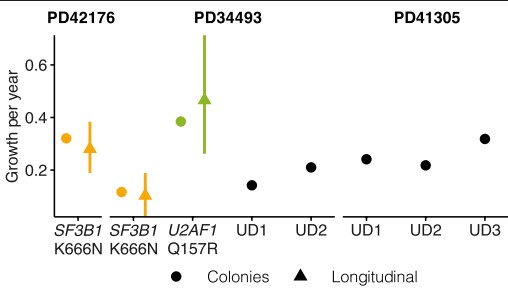

**Extended Data Fig. 6 | Lifelong growth in phylogenetic trees.** Comparison between annual growth derived from phylogenies and growth observed in longitudinal data. For the phylogenies this was obtained by fitting an exponential growth curve to the entire phylodynamic trajectory. For growth rates derived from longitudinal data, error bars represent the 90% HPDI; for growth rates derived from phylogenies (colonies), error bars represent +/− the standard error.

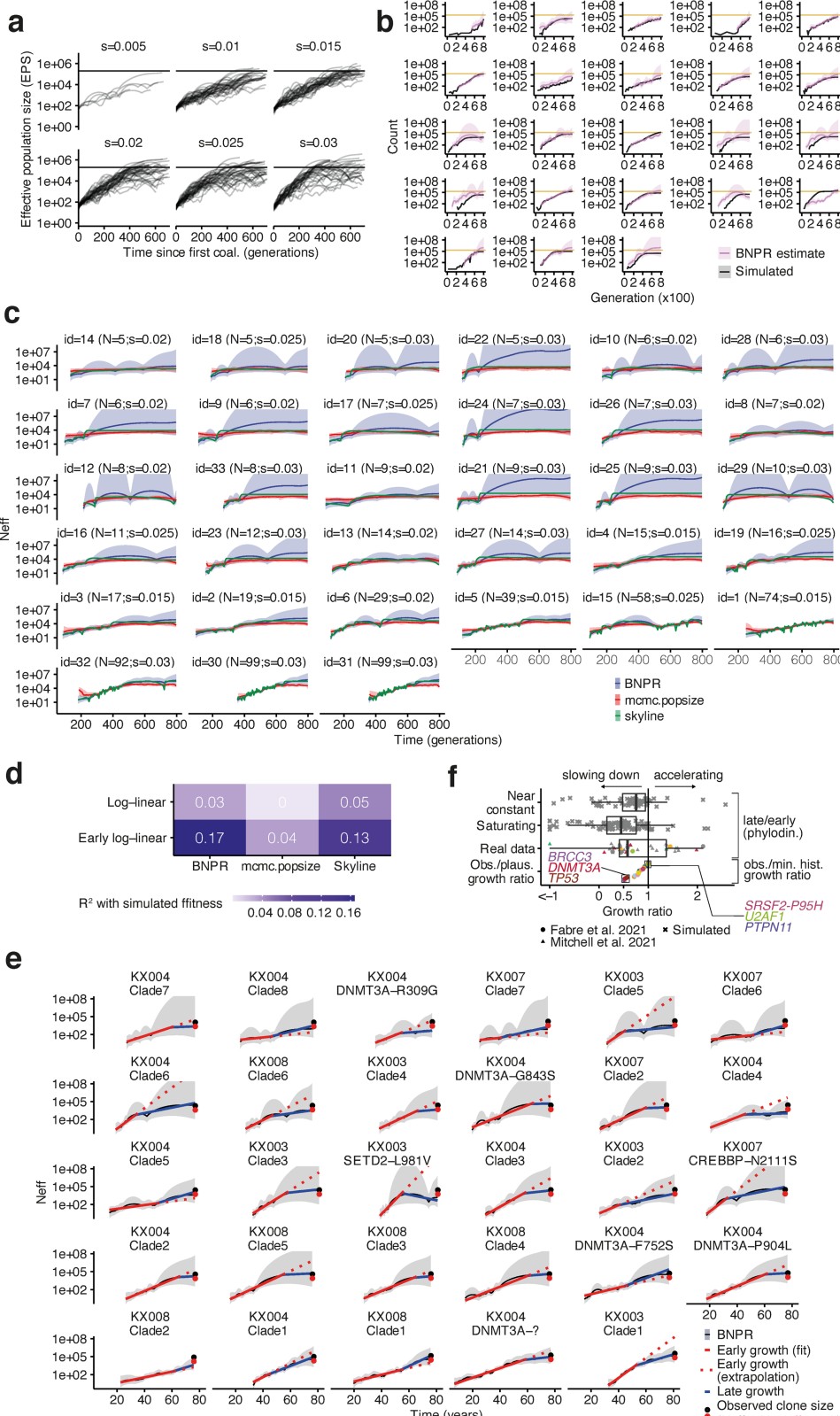

**Extended data Fig. 7** | See next page for caption.

**Extended data Fig. 7 | Examples and consistency of clonal deceleration from simulations and real data. a**, Simulated BNPR trajectories from Wright-Fisher simulations with a fixed population size across 800 generations for a range of fitness effects (0.005, 0.010, 0.015, 0.020, 0.025, 0.030).
**b**, Comparison between Wright-Fisher simulations (grey) and BNPR estimates from phylogenies obtained from these simulations (pink). The horizontal golden line in each plot represents the HSC population carrying capacity (200,000). **c**, Representation of effective population size (Neff) trajectories using three distinct methods (BNPR, mcmc.popsize and skyline; details in the Supplementary Methods) for their estimation across a range of clade sizes and fitness effects. **d**, Quantification of the association between true and inferred fitness values for three distinct methods of Neff estimation.
**e**, Schematic representation of all trajectories from Mitchell et al.[36] and how extrapolating from the initial growth rate leads to the overestimation of the observed clone size (here the observed clone size is obtained by scaling the proportion of tips in a clade by a total Neff of either 200,000 or 1,000,000 HSC x yr). **f**, Quantification of the deceleration effect from real data and simulations (n = 177/n = 37/n = 633 clones detected in simulated phylogenies (top)/haematopoietic phylogenies (middle)/with targeted sequencing (bottom) respectively). The boxes represent the 25th, 50th (median) and 75th percentiles of the data; the whiskers represent the lowest (or highest) datum within 1 interquartile range from the 25th (or 75th) percentile.

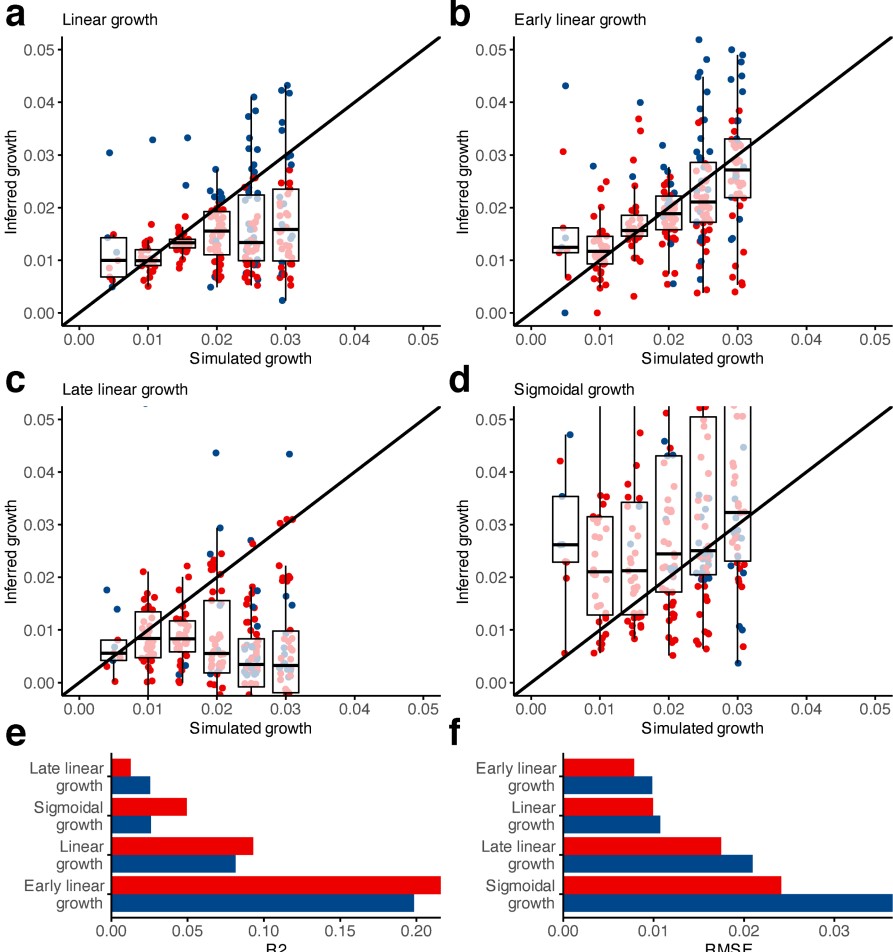

**Extended Data Fig. 8 | Estimation of the true clone fitness from phylodynamic estimation.** Three fits were tested to estimate the true clone fitness from phylodynamic estimation of the population size and these estimates were plotted as a function of the true fitness size (0.005, 0.010, 0.015, 0.020, 0.025 or 0.030). **a**, A log-linear fit; **b**–**c**, A biphasic fit that estimates an early and a late growth rate and a change-point between both and **d**, a sigmoidal fit (n = 241 simulated trajectories). **e**, Coefficient of correlation (R2) for all four inferred coefficients. **f**, Root mean squared error (RMSE) for all four inferred coefficients. In this figure red represents "low variance trajectories" (the average estimated variance for the logarithm of the trajectory is under 5) and blue represents "all trajectories". The boxes in a-d represent the 25th, 50th (median) and 75th percentiles of the data; the whiskers represent the lowest (or highest) datum within 1 interquartile range from the 25th (or 75th) percentile.

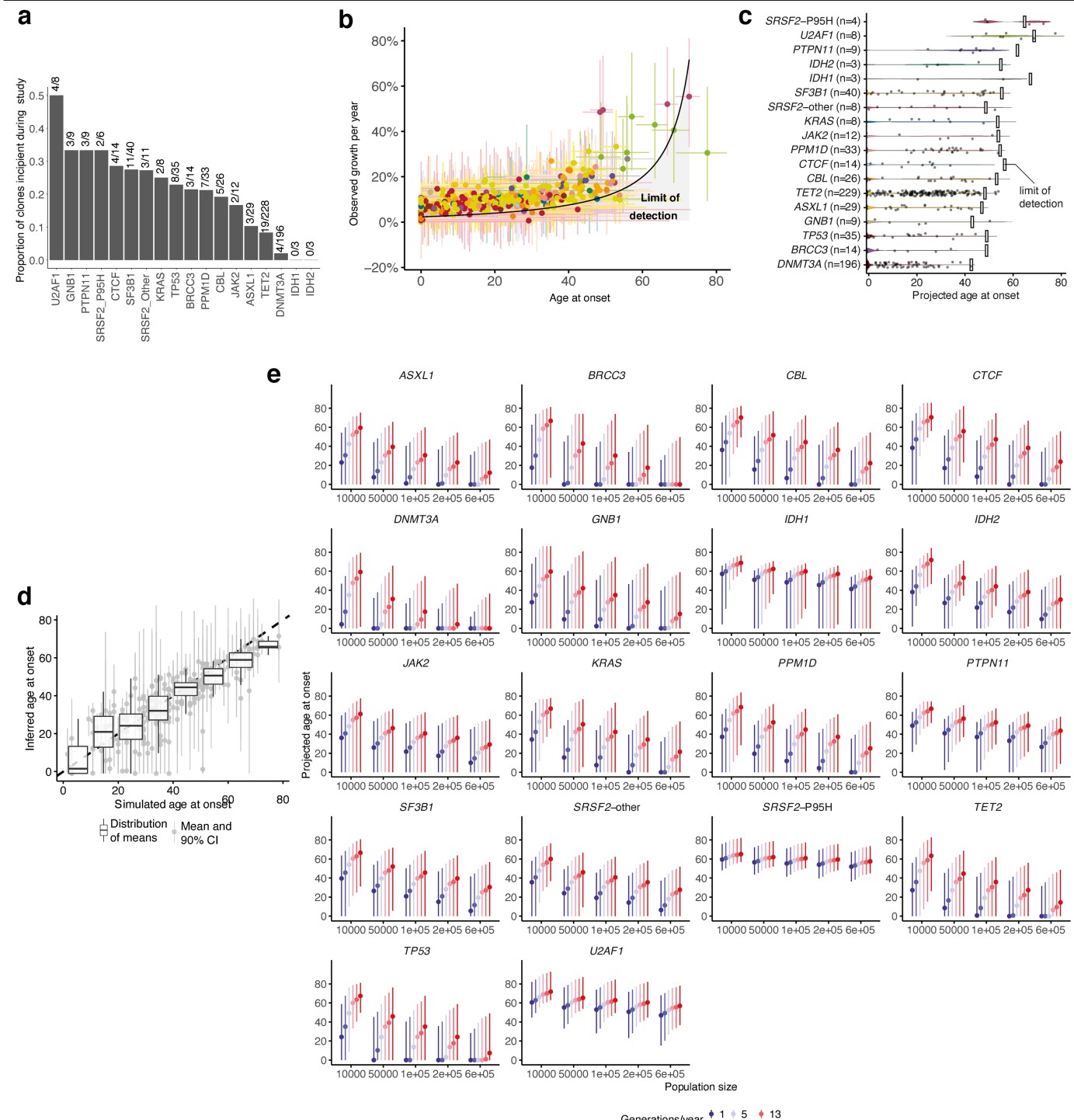

**Extended Data Fig. 9 | Age at clone detection and onset. a**, Proportion of clones driven by different driver mutations that were incipient on-study, ie. undetectable at time-point 1 and detectable by the end-of-study. Absolute numbers are given above each bar. **b**, Relationship between age at onset and observed annual growth rate, with points representing the mean annual growth/median age at onset and intervals representing, respectively, the 90%/95% highest posterior density intervals (HPDI). The black line and grey shaded area represent the theoretical limit of detection at 80 years of age (n = 615 clones). **c**, Violin plot showing the distribution of projected ages at onset for all clones, assuming stable lifelong growth at the same fixed rate we observed during older age. **d**, Association between the age at which clones

appeared in the simulations and the age at clone foundation inferred using our time-series data (R² = 0.75). Boxplots show that, while these estimates may have high variance, the distribution of expected values is close to the true value (n = 250 simulated clones). The boxes represent the 25th, 50th (median) and 75th percentiles of the data; the whiskers represent the lowest (or highest) datum within 1 interquartile range from the 25th (or 75th) percentile. **e**, Sensitivity analysis depicting the median (dot) and the 95% confidence interval of the ages at onset for each gene when considering different population sizes ($10^4$, $5*10^4$, $10^5$, $2*10^5$ and $6*10^5$) and numbers of generations per year (1, 2, 5, 10, 13, 20; n = 615 clones).

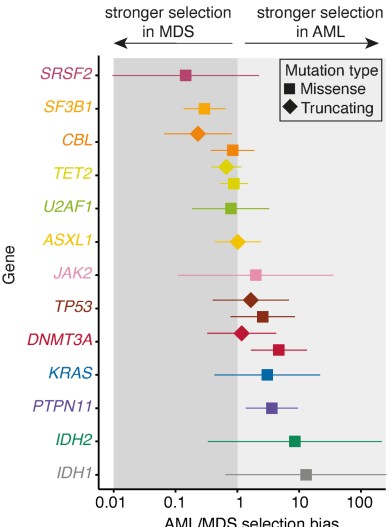

**Extended Data Fig. 10 | Selection in myeloid malignancies. a**, Ratio between AML dN/dS and MDS dN/dS for different genes and mutation types (missense, truncating). If this ratio is >1 there is a bias towards AML, if it is <1 there is a bias towards MDS. Error bars depict 95% CIs.

# Reporting Summary

## Statistics

For all statistical analyses, confirm that the following items are present in the figure legend, table legend, main text, or Methods section.

| n/a | Confirmed | |
|---|---|---|
| ☐ | ☒ | The exact sample size (*n*) for each experimental group/condition, given as a discrete number and unit of measurement |
| ☐ | ☒ | A statement on whether measurements were taken from distinct samples or whether the same sample was measured repeatedly |
| ☐ | ☒ | The statistical test(s) used AND whether they are one- or two-sided<br>*Only common tests should be described solely by name; describe more complex techniques in the Methods section.* |
| ☐ | ☒ | A description of all covariates tested |
| ☐ | ☒ | A description of any assumptions or corrections, such as tests of normality and adjustment for multiple comparisons |
| ☐ | ☒ | A full description of the statistical parameters including central tendency (e.g. means) or other basic estimates (e.g. regression coefficient) AND variation (e.g. standard deviation) or associated estimates of uncertainty (e.g. confidence intervals) |
| ☐ | ☒ | For null hypothesis testing, the test statistic (e.g. *F*, *t*, *r*) with confidence intervals, effect sizes, degrees of freedom and *P* value noted<br>*Give P values as exact values whenever suitable.* |
| ☐ | ☒ | For Bayesian analysis, information on the choice of priors and Markov chain Monte Carlo settings |
| ☐ | ☒ | For hierarchical and complex designs, identification of the appropriate level for tests and full reporting of outcomes |
| ☐ | ☒ | Estimates of effect sizes (e.g. Cohen's *d*, Pearson's *r*), indicating how they were calculated |

*Our web collection on statistics for biologists contains articles on many of the points above.*

## Software and code

Policy information about availability of computer code

| | |
|---|---|
| Data collection | Image processing from sequencing data using standard Illumina NovaSeq and HiSeq pipelines. |
| Data analysis | Read alignment was performed using BWA-MEM (v0.7.17). Somatic mutation calling and VAF quantification was performed using Shearwater (v.1.21.5), CaVEMan (v.1.11.2), Pindel (v.2.2) and the in-house script vafCorrect (https://github.com/cancerit/vafCorrect). Allele counts at recurrent mutation hotspots were verified using an in-house script (github.com/cancerit/allelecount). Copy number aberrations in blood colonies WGS were determined using ASCAT (v4.2.1) and BRASS (https://github.com/cancerit/BRASS). Phylogenetic trees were derived using MPBoot (version 1.1.0 for Linux (http://www.iqtree.org/mpboot)) and treemut (https://github.com/NickWilliamsSanger/treemut) and validated using SCITE (https://gitlab.com/jahnka/SCITE). Population size trajectories were determined using Bayesian nonparametric phylodynamic reconstructions (BNPR) implemented in the phylodyn (v0.9.2) package for R. dNdScv (v.0.0.1.0) was used to quantify selection for somatic mutations. Wright-Fisher simulations of the haematopoietic system were performed using clonex (https://github.com/gerstung-lab/clonex).<br>All statistical analyses used the software R (v.3.6.3). All R files used for the longitudinal and phylodynamic modelling and validation are publicly available at https://github.com/josegcpa/clonal_dynamics. All files used for the construction of phylogenetic trees are publicly available at https://github.com/margaretefabre/Clonal_dynamics.<br>A list of all the R packages explicitly used (excluding package prerequisites) and their respective versions (in brackets) is provided here: MASS (v7.3), ape (v5.4), bayesplot (v1.7.2), car (v3.0), castor (v1.6.4), coda (v0.19), colorspace (v1.4), cowplot (v1.1.0), dendextend (v1.14.0), extraDistr (v1.9.1), ggforce (v0.3.2), ggpubr (v0.4.0), ggrepel (v0.8.2), ggridges (v0.5.2), ggsci (v2.9), ggsignif (v0.6.0), ggtree (v2.0.4), greta (v0.3.1), gridExtra (v2.3), gtools (v3.8.2), nlme (v3.1), openxlsx (v4.2.2), phangorn (v2.5.5), phylodyn (v0.9.2), reghelper (v1.0.1), rreticulate (v1.16), scatterpie (v0.1.5), stringr (v1.4.0), survival (v3.2), survminer (v0.4.8), tensorflow (v2.2.0), tidyverse (v1.3.0), zoo (v1.8), deepSNV (v1.21.5) and dNdScv (v.0.0.1.0). |

For manuscripts utilizing custom algorithms or software that are central to the research but not yet described in published literature, software must be made available to editors and reviewers. We strongly encourage code deposition in a community repository (e.g. GitHub). See the Nature Portfolio guidelines for submitting code & software for further information.

## Data

Policy information about availability of data

All manuscripts must include a data availability statement. This statement should provide the following information, where applicable:

- Accession codes, unique identifiers, or web links for publicly available datasets
- A description of any restrictions on data availability
- For clinical datasets or third party data, please ensure that the statement adheres to our policy

The data files necessary to run the analysis in https://github.com/josegcpa/clonal_dynamics are freely available at https://doi.org/10.6084/m9.figshare.15029118. All sequencing data have been deposited in the European Genome-phenome Archive (EGA) (https://www.ebi.ac.uk/ega/). Targeted sequencing data have been deposited with EGA accession numbers EGAD00001007682 and EGAD00001007683; WGS data have been deposited with accession number EGAD00001007684. Data from the EGA are accessible for research use only to all bona fide researchers, as assessed by the Data Access Committee (https://www.ebi.ac.uk/ega/about/access). Data can be accessed by registering for an EGA account and contacting the Data Access Committee.

AML datasets were retrieved from Papaemmanuil et al, NEJM, 2016 (Ref...)
MDS datasets were retrieved from Papaemmanul et al, Blood 2013 (Ref...)
Phylogenetic data from the Mitchell et al companion paper was kindly shared with us from the authors of the paper.

# Field-specific reporting

Please select the one below that is the best fit for your research. If you are not sure, read the appropriate sections before making your selection.

☒ Life sciences ☐ Behavioural & social sciences ☐ Ecological, evolutionary & environmental sciences

For a reference copy of the document with all sections, see nature.com/documents/nr-reporting-summary-flat.pdf

# Life sciences study design

All studies must disclose on these points even when the disclosure is negative.

| | |
|---|---|
| Sample size | To enable us to derive robust estimates of clonal dynamics over time we wanted to study a sufficient number of individuals with clonal haematopoiesis driven by each of its 9 most common driver genes (DNMT3A, TET2, ASXL1, SF3B1, TP53, SRSF2, PPM1D, JAK2 and U2AF1). Our expectation was that clones driven by mutations in the same gene would behave relatively similarly, but the extent of similarity was not known at the outset of the study. We therefore aimed to capture at least 6-8 cases of CH driven by each of these genes. As U2AF1 is the least commonly mutated of these genes (approximately 2-3% of individuals in the studied age range), this was the gene that determined our sample size of 385 participants, which gave an expectation that we would identify approximately 7-12 individuals with U2AF1-driven clonal haematopoiesis. In the end we identified 8 such cases, whilst numbers of individuals with mutations in each of the other 8 genes were greater. |
| Data exclusions | We excluded DNA sequencing reads that did not meet widely accepted quality metrics. We also excluded 16 single cell colony-derived WGS data that did not meet set quality criteria. |
| Replication | This is the first study to sequence serially obtained samples in order to study the dynamic behaviour of clonal haematopoiesis. As such, no comparable datasets are available for replication. |
| Randomization | N/A - there was no intervention or treatment studied. |
| Blinding | N/A - there was no intervention or treatment studied. |

# Reporting for specific materials, systems and methods

We require information from authors about some types of materials, experimental systems and methods used in many studies. Here, indicate whether each material, system or method listed is relevant to your study. If you are not sure if a list item applies to your research, read the appropriate section before selecting a response.

## Materials & experimental systems

| n/a | Involved in the study |
|-----|------------------------|
| ☒ ☐ | Antibodies |
| ☒ ☐ | Eukaryotic cell lines |
| ☒ ☐ | Palaeontology and archaeology |
| ☒ ☐ | Animals and other organisms |
| ☐ ☒ | Human research participants |
| ☒ ☐ | Clinical data |
| ☒ ☐ | Dual use research of concern |

## Methods

| n/a | Involved in the study |
|-----|------------------------|
| ☒ ☐ | ChIP-seq |
| ☒ ☐ | Flow cytometry |
| ☒ ☐ | MRI-based neuroimaging |

# Human research participants

Policy information about studies involving human research participants

| | |
|---|---|
| Population characteristics | The 385 study participants (199 women) were aged 54-93 years (median 69.3) at study entry and were sampled up to 5 times (median 4) over 3.2-16 years (median 12.9 years). The participants had no history of haematological malignancy, but were otherwise unselected. |
| Recruitment | Participants were recruited as part of the Sardinia study. The study recruited unselected Sardinians aged 14-102 years with the initial sample cohort of >6000 people including over 62% of the eligible population living in the catchment region in Ogliastra. Recruitment of the individuals in the cohort occurred prior to the large studies by us and others reporting the frequency of clonal haematopoiesis and the commonly mutated driver genes in 2014-2015. Our study investigated 385 individuals aged 54 years or older, who were randomly selected from Sardinia study participants without any identifiable selection bias (see - https://sardinia.nia.nih.gov/). Individuals who developed a haematological malignancy before or after recruitment were excluded from the present study. |
| Ethics oversight | Ethical permission for this study was granted by The East of England (Essex) Research Ethics Committee (REC reference 15/EE/0327). All applicants signed informed consent, which is now stated clearly in the manuscript. |

Note that full information on the approval of the study protocol must also be provided in the manuscript.

