## [Peer Review File · Nature]

Manuscript Title: The longitudinal dynamics and natural history of clonal haematopoiesis

Reviewer Comments & Author Rebuttals

Reviewer Reports on the Initial Version:

Referees' comments:

Referee #1 (Remarks to the Author):

Fabre et al identified clonal haematopoiesis clones in ~400 subjects and tracked those clones over a median time of 13 years. They found significantly different growth rates of these clones depending on the mutated driver genes. The growth rate variance within each driver gene was relatively low. Additionally, the authors analyzed whole genome sequencing data from 1,731 haematopoietic colonies from 7 subjects and found that DNMT3A-mutant clones tend to expand early in life while splicing gene mutant clones tend to expand later in life while TET2-mutant clones harbored relatively stable growth rates. Faster growing clones had a higher risk of malignant progression. It is also interesting to see how TP53 mutant clones with missense mutations grew at different rates than those with truncating mutations. Moreover, the authors observed that many clones grew much faster in earlier life than at an older age and that mutations associated with faster clonal haematopoiesis growth are also associated with a higher risk of progression to AML. In summary, this is an exciting body of work with many new insights based on a unique dataset.

Line 54: I don't think that recent evolutionary models "propose" that each specific mutation carries a fixed fitness advantage. These models "assume" a fixed fitness advantage due to lack of comprehensive longitudinal data as well as tractability of the mathematical models. I don't think any mathematical modeler thinks that fitness advantages are independent of the changing microenvironment.

Ps: Very much appreciate that the authors put the figures and their captions together at the right position.

Referee #2 (Remarks to the Author):

In this manuscript, Fabre et al address an important set of questions with regard to when clonal haematopoiesis develops, what the clonal growth dynamics are over time, and how this relates to malignant progression. Strengths of the paper include longitudinal tracking of clones over an extended period of time (median of 13 years) and integration with phylogenetic analyses in ways that enabled interesting and robust conclusions. Their results are of interest for many reasons including (i) the finding that certain mutations appear to drive faster growth of clones and increase

risk of progression to malignancy is one with potential clinical significance; (ii) the ability to predict future clonal growth trajectories may have clinically significant implications; and (iii) their findings are able to provide an explanation for the varying prevalence of CH mutations with age based on differential growth dynamics (i.e., as with differential prevalence of DNMT3A and TET2 mutations in older age; rarity of certain mutations in younger individuals).

- Could the authors clarify whether the DNA samples that they analyzed in their longitudinal analyses were from PBMCs or whole blood DNA (as this influences interpretation of VAF/ clone size)?
- The authors state, based on their results in Figure 1d, that the prevalence of DNMT3A mutations showed no significant relationship with age overall. This result seems to be driven mostly by the reduction in prevalence in individuals > 85 years of age, and there does seem to be an age-related increase in prevalence in individuals < 85 years of age. While they present compelling evidence later in the paper as to the likely reason for this (that DNMT3A mutations seem to arise and grow faster earlier in life), can they exclude the possibility that older (>85) individuals in their study may appear to have a lower frequency of DNMT3A mutations because those carrying the mutation may have been excluded as a result of having already developed malignancy or being more likely to have died?
- In Figure 2a, the authors present examples of fitted exponential growth with mutations at common hotspots. Some of these (e.g., U2AF1 Q157R in subject PD41345) appears to involve just two time points. Can the authors present an analysis of the goodness-of-fit of their model broken down by the number of samplings (i.e., at least 3, at least 4, and 5)? Although the expectation would be that the model would indeed still a good fit given that the median number of samplings was 4, this could help clarify where most of the signal is coming from.
- In their model for determining clonal growth rate (which included the mutated gene, specific mutation, and mutation type), did the authors consider include clone size at the time of initial sampling? Could this allow for prediction of future clonal trajectories with greater accuracy?
- How do the authors reconcile their finding described in lines 186-187, that “TET2-mutant clones, which grew faster in older individuals” with the statement in the abstract, that “TET2-mutant clones showed minimal age-dependency”?
- What were the criteria used by the authors to choose the specific 3 individuals on whom they performed their phylogenetic analyses (in Fig 3)?
- Given overall growth rate of SRSF2-P95H clones (> 50%/ year), how do the authors reconcile the finding in Figure 3c, that this mutation appears to have arisen in this individual early on life (teens) yet is present in only 1 of 96 colonies sequenced at age 73 in this individual? If this is potentially because of competition with the (multiple) clones with unknown drivers in this individual, what does that say about the need to account for such clones in being able to accurately model the growth rate of clones with known drivers?

- While the authors provide compelling evidence that many clones are likely to have grown more rapidly early in life compared to the observed rate in old age, are there DNA samples available from younger individuals (i.e., even if the same individuals analysed in this study do not have earlier DNA samples) in whom the robustness of this conclusion can be further evaluated empirically?
- Is their analysis in Figure 5a controlled for starting clone size?

Referee #3 (Remarks to the Author):

Primary uniqueness of this article is the collection of temporal variant frequencies in 56 genes collected over 13 years in 385 older individuals. This allowed for a study of longitudinal dynamics of clonal haematopoiesis (CH), with a few observations made and inferences drawn. Because the individuals were relatively old, some variants seeded lineages that grow faster than others (CH), including those in some key genes involved in CH-related cancers. Overall, however, I found this manuscript to be mostly descriptive with no scintillating discovery. The analysis of generic variation observed, while competent, is light on novelty.

Overall, this work is more suitable for a specialist human genetics journal. I have a few other comments, which are given below.

- (1) Authors do not discuss any functional genomics aspects of variants. For example, truncating and missense mutations show similar patterns and they are all heterozygotes. Why are the patterns the same? What about their haplosufficiency, dominance, and loss/gain-of-function?
- (2) Page 15-16, lines 448-449: The assumption of the VAF to be representative of the prevalence of a single clone needs to be re-evaluated. For example, in Figure 3b, clone 3 with Unknown driver 2 produced clone 2 with U2AF1 mutation. This means that the VAF of Unknown driver 2 represents clones 3 and 2. The impact of this violation on the conclusion should be discussed, unless I am mistaken.
- (3) Also, line 448 mentions that phasing different mutations into specific clones was impossible. However, there are many computational methods to phase different mutations into specific clones when VAF is available from multiple timepoints or samples (e.g., PyClone and CITUP). Many studies with deep sequencing have reported mutation clusters. Would that information be not helpful to even reconstruct better mutational history?
- (4) For the single cell sequencing data, it is unclear why MPBoot was sufficient. This regular approach is not going to be effective if there are many missing bases and erroneous mutation assignments. There is a critical need to provide information on data quality and the robustness of the inferences to the choice of method (e.g., MP vs SCITE).
- (5) The accuracy of the inferred phylogenetic tree from single cell sequencing data are likely tentative, as the MP method cannot deal with data containing many missing data and incorrect base assignments. Methods designed designed for low-quality and single-cell sequencing data should be used (e.g., SCITE).
- (6) In fact, the quality of sequence data should be reported, e.g., rate of unknown base assignments and sensitivity of variant calling (page 19 lines 579-580).

(7) Page 19 lines 582-584: The time axis was scaled linearly from the conception (the root of the tree) to the age of sampling (the tips), and constant mutation rates were assumed to generate a timeline. It is unlikely that mutation rate was constant from the time of the conception to the age of sampling. In contrast, there may be many phases of clock ticking at different rates.

(8) Did the dN/dS ratio the same across time in variation data and in sc-Seq data, excluding the CH clones?

Author Rebuttals to Initial Comments:

Response to reviewers' comments on the manuscript entitled
"The longitudinal dynamics and natural history of clonal
haematopoiesis"

We thank our reviewers for their time and helpful critique of our manuscript. We are delighted with their positive overall assessment of our work and its impact, and found their comments both instructive and insightful. We have addressed these by generating new data and performing extensive additional analyses of data generated by us and others. We believe that, in addressing these comments, we have improved the robustness and clarity of our findings. We hope that our reviewers concur with this view and agree that our manuscript makes important novel contributions to our understanding of the lifelong behaviour of different types of CH and their links to haematological malignancies. Our findings will be of deep interest to those studying somatic clonal evolution, ageing and cancer, and of general interest to the wider scientific community.

Below, we provide a point-by-point response to their comments and refer our reviewers to the relevant changes/additions to our manuscript. Reviewer comments are in **black**, with our responses in **blue** and actions that we have undertaken for the revision in **red**. Within our revised manuscript, changes/additions made in response to reviewers' comments are also highlighted in **red** and are referred to by line number in the following responses.

Referee #1 (Remarks to the Author):

- Fabre et al identified clonal haematopoiesis clones in ~400 subjects and tracked those clones over a median time of 13 years. They found significantly different growth rates of these clones depending on the mutated driver genes. The growth rate variance within each driver gene was relatively low. Additionally, the authors analyzed whole genome sequencing data from 1,731 haematopoietic colonies from 7 subjects and found that DNMT3A-mutant clones tend to expand early in life while splicing gene mutant clones tend to expand later in life while TET2-mutant clones harbored relatively stable growth rates. Faster growing clones had a higher risk of malignant progression. It is also interesting to see how TP53 mutant clones with missense mutations grew at different rates than those with truncating mutations. Moreover, the authors observed that many clones grew much faster in earlier life than at an older age and that mutations associated with faster clonal haematopoiesis growth are also associated with a higher risk of progression to AML. In summary, this is an exciting body of work with many new insights based on a unique dataset.

We thank referee #1 for the positive assessment of our work.

- Line 54: I don't think that recent evolutionary models "propose" that each specific mutation carries a fixed fitness advantage. These models "assume" a fixed fitness advantage due to lack of comprehensive longitudinal data as well as tractability of the mathematical models. I don't think any mathematical modeler thinks that fitness advantages are independent of the changing microenvironment.

We agree with this comment. We have modified the relevant text to clarify this point and avoid confusion (Lines 54-55).

- Ps: Very much appreciate that the authors put the figures and their captions together at the right position.

Referee #2 (Remarks to the Author):

- In this manuscript, Fabre et al address an important set of questions with regard to when clonal hematopoiesis develops, what the clonal growth dynamics are over time, and how this relates to malignant progression. Strengths of the paper include longitudinal tracking of clones over an extended period of time (median of 13 years) and integration with phylogenetic analyses in ways that enabled interesting and robust conclusions. Their results are of interest for many reasons including (i) the finding that certain mutations appear to drive faster growth of clones and increase risk of progression to malignancy is one with potential clinical significance; (ii) the ability to predict future clonal growth trajectories may have clinically significant implications; and (iii) their findings are able to provide an explanation for the varying prevalence of CH mutations with age based on differential growth dynamics (i.e., as with differential prevalence of DNMT3A and TET2 mutations in older age; rarity of certain mutations in younger individuals).

We thank referee #2 for appreciating the relevance of our work and for highlighting several salient and interesting aspects.

- Could the authors clarify whether the DNA samples that they analyzed in their longitudinal analyses were from PBMCs or whole blood DNA (as this influences interpretation of VAF/clone size)?

We apologise for not making this clear, as it is pertinent to our work. DNA was extracted from whole blood and **this is now stated in Methods (Line 420)**. Although whole blood DNA has been used routinely in studies of CH, we appreciate that CH mutations in blood are primarily harboured by myeloid cells and took additional measures to account for this. *Firstly*, as a minority of myeloid drivers are also found in some lymphoid malignancies, we took steps to ensure that the mutations we detected were indeed myeloid-associated. In particular, we did not include any individual with a persistently elevated peripheral blood lymphocyte count (defined as $\geq 5 \times 10^9/L$ on two consecutive phases of sample collection, or $\geq 5 \times 10^9/L$ at the final phase).

Secondly, and in response to this comment, we took the opportunity to test the robustness of using whole blood DNA in the 11 individuals in our study that were re-sampled for the purpose of validating our ability to predict future clonal growth (Extended Data Fig. 3f-h). As we had already purified granulocytes from these individuals, we sequenced granulocyte DNA in parallel with whole blood DNA (from the same blood sample), using the targeted protocol described in Methods (p.15). Comparison of Variant Allele Fractions (VAFs) in granulocytes versus whole blood showed an excellent correlation (Figure R1; simple linear regression: $R^2 = 0.94$, $p = 2.3 \times 10^{-9}$).

Figure R1: Comparison of VAFs measured in DNA extracted from granulocytes versus whole blood

This confirms that VAF measured in whole blood is a faithful reflection of VAF in the myeloid compartment, and supports the use of whole blood VAF trajectories to infer the dynamics of clonal haematopoiesis.

- The authors state, based on their results in Figure 1d, that the prevalence of DNMT3A mutations showed no significant relationship with age overall. This result seems to be driven mostly by the reduction in prevalence in individuals > 85 years of age, and there does seem to be an age-related increase in prevalence in individuals < 85 years of age. While they present compelling evidence later in the paper as to the likely reason for this (that DNMT3A mutations seem to arise and grow faster earlier in life), can they exclude the possibility that older (>85) individuals in their study may appear to have a lower frequency of DNMT3A mutations because those carrying the mutation may have been excluded as a result of having already developed malignancy or being more likely to have died?

We thank the referee for raising this valid question. However, we do not believe that the lower prevalence of *DNMT3A* mutations in very elderly individuals is due to sampling bias, for reasons outlined below.

First, the referee correctly points out that we excluded any individual with haematological malignancy, including both those with a historical diagnosis and those diagnosed on-study. Crucially, the incidence of haematological malignancy is so low that it could not have materially impacted our observed prevalence of *DNMT3A*-mutant CH. Among the entire SardiNIA cohort, only 22/7816 (0.28%) individuals (age range 18 - 107 years, median 53.5) had a past history of haematological malignancy or developed such disease during the ~15 years of follow-up. This is in keeping with the low incidence of haematological malignancies reported in epidemiological studies [1]. Also, it is notable that clones with mutations in genes other than *DNMT3A* are also susceptible to malignant progression, and we and others have previously shown that several of these mutant genes (e.g.

SRSF2, *SF3B1*, *TET2*, *ASXL1*, *TP53*) confer a higher risk of progression than *DNMT3A* [2], yet we see no comparable fall in their prevalence in extreme old age.

To address the question of whether individuals with *DNMT3A* mutations might have been excluded from our study because they were more likely to have died, we have now looked for evidence that *DNMT3A* mutations impact survival. First, within our longitudinal cohort of 385 elderly individuals, we obtained survival data and used the *survival* package in R [3] to construct a Cox proportional hazards model. We found no difference in survival between individuals with vs without *DNMT3A* mutations, while controlling for age (at last time-point), sex, number of clones, size of largest clone, diabetes status and smoking history (Figure R2). Compared to individuals with no CH, this lack of difference in survival was true both for those with CH driven by *DNMT3A* mutation alone (HR 1.35 (0.44-4.1), $p=0.6$) and for those with CH clones driven by mutations in *DNMT3A* and other genes (HR 0.61 (0.19-2.0), $p=0.41$) (Figure R2).

Figure R2. Kaplan-Meier plots showing survival of 385 individuals in our longitudinal cohort, stratified by CH mutation status.

We now include reference in our manuscript to the fact that changes in driver gene prevalence with age could not have resulted from exclusion of individuals with haematological malignancies, as the incidence of these in the cohort was so low (Lines 101-104). In the interest of clarity we do not refer directly to survival, as there is no evidence that *DNMT3A* mutations impact this.

- In Figure 2a, the authors present examples of fitted exponential growth with mutations at common hotspots. Some of these (e.g., U2AF1 Q157R in subject PD41345) appears to involve just two time points. Can the authors present an analysis of the goodness-of-fit of

their model broken down by the number of samplings (i.e., at least 3, at least 4, and 5)? Although the expectation would be that the model would indeed still be a good fit given that the median number of samplings was 4, this could help clarify where most of the signal is coming from.

This is a relevant point. We assessed the goodness-of-fit of our logistic regression model of clonal trajectories using the number of outlying data-points. We defined ‘explained’ trajectories as those with no outliers. We did not assess how the number of available time-points might impact the fraction of explained trajectories, but have now done so. Figure R3 shows that increasing the number of timepoints beyond 3 has minimal impact on goodness-of-fit. While trajectories with only two time-points are guaranteed to have perfect fits, only 11/385 individuals (26/685 trajectories) had such few time-points (Extended Data Fig. 1a). The impact of excluding these 11 individuals is marginal; without these, we find the proportion of trajectories showing fixed-rate growth (ie. no outliers) drops from 92.4% to 92.1%. Even among trajectories with 5 time-points, overall 91% have no outliers.

Figure R3: Fraction of explained trajectories by number of timepoints available

Figure R3 is now included in our manuscript as Extended Data Fig. 3e, and is referred to in the text of the manuscript (Line 120).

We also note no major bias within each gene regarding the fraction of explained trajectories and the number of available time-points (Figure R4), which is important for interpretation of the per-gene explained trajectories presented in Figure 2b.

Figure R4. Fraction of explained trajectories by number of timepoints available, stratified by gene (top 6 most prevalent driver genes are shown).

- In their model for determining clonal growth rate (which included the mutated gene, specific mutation, and mutation type), did the authors consider including clone size at the time of initial sampling? Could this allow for prediction of future clonal trajectories with greater accuracy?

We thank the reviewer for this insightful comment. Our model includes an offset term, which accounts implicitly for different clone sizes (VAFs) and ages observed at study entry. In addition, as well as capturing genetic effects on growth (mutant gene, site and mutation type), our model also quantifies a residual term that captures growth not explained by these genetic effects, i.e. how much an individual clone's growth rate differs from the average growth rate of all clones bearing the same driver. We refer to this residual term as 'unknown-cause growth' and have not previously tested if starting clone size affects this. We now query this directly by assessing whether the unknown-cause growth effect of each clone is associated with initial clone size. We observe a small negative association, with VAF accounting for 6.2% of the variance in unknown-cause effect ($R^2 = 0.062$; $p\text{-value} = 2.42 \times 10^{-9}$; Figure R5).

Figure R5. Association between VAF at study entry and unknown-cause growth of individual clones.

Importantly, since unknown-cause growth is itself generally a minor contributor to overall clonal dynamics (Fig. 2d), the influence of clone size at study entry is likely to be small. The possible reasons for the small negative association between starting clone size and growth rate are speculative, but might relate to larger clones being subject to stronger deceleration as the competitive oligoclonal landscape steadily saturates haematopoiesis in old age (Fig. 4).

Since the small impact of initial clone size is already contained within the ‘unknown-cause growth’ component of our model, our trajectory predictions already account for this variable.

We now include Figure R5 in our manuscript (as Extended Data Fig. 4e) and refer to the association between initial clone size and unknown-cause growth in the text (Lines 187-188).

- How do the authors reconcile their finding described in lines 186-187, that “TET2-mutant clones, which grew faster in older individuals” with the statement in the abstract, that “TET2-mutant clones showed minimal age-dependency”?

Thank you for pointing out this ambiguity in the wording of our manuscript. Our use of the term “minimal age-dependency” referred to the ability of TET2 mutations to initiate clones throughout life. We have now made this clear by altering the wording of the abstract (Lines 36-37). Separately, we noticed that unknown-cause growth was positively associated with age, i.e. within our elderly cohort TET2-mutant clones tended to grow faster in the very old (Extended Data Fig. 4g).

- What were the criteria used by the authors to choose the specific 3 individuals on whom they performed their phylogenetic analyses (in Fig 3)?

We chose individuals with clones driven by splicing gene mutations, as a previous report [4] suggested a sharp increase in their prevalence late in life. We have now made this clearer in the

manuscript (Lines 197-198). In light of this, we wanted to investigate whether this was due to late clonal initiation (of new clones) or sudden acceleration in older age of pre-existing clones (that were previously not detectable by standard sequencing of blood DNA).

Also, we were aware that the accompanying study by Mitchell *et al.* involved sequencing of samples that happened to have *DNMT3A*-mutant clones and were happy to share their data with us for the specific task of studying lifelong clonal behaviour. In this way, we would be able to investigate the two opposite types of behaviour (early growth for *DNMT3A*-mutant CH and late growth for splicing gene-mutant CH).

- Given overall growth rate of *SRSF2*-P95H clones (> 50%/ year), how do the authors reconcile the finding in Figure 3c, that this mutation appears to have arisen in this individual early on life (teens) yet is present in only 1 of 96 colonies sequenced at age 73 in this individual? If this is potentially because of competition with the (multiple) clones with unknown drivers in this individual, what does that say about the need to account for such clones in being able to accurately model the growth rate of clones with known drivers?

We thank the reviewer for this comment, however the phylogenetic tree in Figure 3c unfortunately does not allow us to accurately time the onset of the *SRSF2*-P95H clone, as it is consistent with an age of onset between 13 and the age at sampling (73 years), i.e. anywhere along the long phylogeny branch that includes this mutation. Nevertheless, we agree that, in advanced old age, haematopoiesis does become increasingly oligoclonal (as also revealed by Mitchell *et al.* [5]), such that expanding clones might begin to saturate haematopoiesis and limit each other's growth. Indeed, as the reviewer suggests, such competition may influence the clonal behaviours we observe across our cohort. If competition does affect growth rates there are two possible ways in which it may operate. First, if competition was uniform across individuals, this effect would be captured within the driver growth rate in our longitudinal model. Alternatively, if the impact of competition differed between individuals, this would be captured by our "unknown cause growth effect". Therefore, any potential impact of competition would be captured by our model.

We now include more detail on the *SRSF2*-P95H mutant colony in our Results (Lines 208-210) to avoid confusion about its age of onset. Also, we explain how the unknown cause growth effect can capture interclonal competition as well as other factors that may affect clonal growth (Lines 480-482).

- While the authors provide compelling evidence that many clones are likely to have grown more rapidly early in life compared to the observed rate in old age, are there DNA samples available from younger individuals (i.e., even if the same individuals analysed in this study do not have earlier DNA samples) in whom the robustness of this conclusion can be further evaluated empirically?

We agree that this would provide further validation of our conclusions. However, we do not have access to any collection of blood DNA from serially sampled young individuals to address this question. Also, the rarity of CH in the young would require that large numbers are screened to identify sufficient cases of CH for analysis. Therefore, we are not able to directly study serial longitudinal samples from young individuals at present.

Instead, to address this comment we now compare our findings with those of Watson *et al* [6], who used mathematical modelling and evolutionary theory to derive growth rates/coefficients from single time point data in younger individuals. We focus on *DNMT3A*-mutant CH, as this exemplifies the phenomenon in question (i.e. rapid growth in early life followed by deceleration in old age).

Watson *et al* applied population genetic theory to analyse cross-sectional VAF spectra from ~50,000 individuals and infer the fitness of individual CH driver mutations [6]. Two important assumptions in this study were that (i) mutation fitness was constant from the time of clone foundation to the time of sampling, and (ii) that the propensity for a mutation to found a clone was constant throughout life. Therefore, their fitness estimates reflect each mutation's *average* growth rate from mutation acquisition to blood sampling. As the mean age of the 50,000 individuals studied was 55 years (SD 11.4 years), their fitness estimates reflect average clonal growth prior to this age. In comparison, the growth rates we measured in our longitudinal study relate solely to old age, covering, on average, the period between the ages of 69-81 years (median ages at the start and end of study). Thus, the findings of Watson *et al* reflect average clonal growth in earlier life, and those of our longitudinal study reflect clonal growth in later life.

To compare our data to Watson *et al*, we focussed on the mutant sites in *DNMT3A* for which both of our studies had derived site-specific fitness effects, namely R882C, R882H, R736C and R736H. This confirmed that the expansion of clones associated with each of these mutations was substantially faster in younger (Watson *et al* [7]) vs older (our study) individuals (Table R1).

Gene_Mutation	Annual growth in older age (our study) (%/y)		Annual growth in younger age (Watson et al) (%/y) [7]	
	Rate	95% CI	Rate	95% CI
DNMT3A_R882C	1.8	(-6.8 - 11.5)	18.7	(18.2 - 19.4)
DNMT3A_R882H	5.5	(0.7 - 10.9)	14.8	(14.1 - 15.7)
DNMT3A_R736H	10.3	(3.2 - 17.6)	14.1	(13.2 - 15.4)
DNMT3A_R736C	7.3	(0.7 - 14.1)	12.3	(11.6 - 13.4)

Table R1. Annual growth rates associated with *DNMT3A* hotspots in younger (Watson *et al.*) vs older (our study) individuals.

This comparison with the study by Watson *et al* adds further weight to our finding that many clones, particularly those driven by mutant *DNMT3A*, grow more rapidly in early life compared to their growth rate in old age. Also, in addition to evidence which is derived from (i) retrograde extrapolation of our time-series data and (ii) modelling of clonal dynamics from phylogenetic trees,

two additional findings of our study add further support to the premise that clonal growth behaviour often changes over a lifetime:

- Almost all *DNMT3A*-mutant clones were already detectable by the study onset, with only 4/196 (2%) ‘incipient’ on-study (Extended Data Figure 9a). This is compatible with the notion that these clones tend to initiate early in life and grow slowly in old age. This contrasts with the behaviour of clones driven by mutations in genes such as *TET2*, where 19/228 (8%) were undetectable at the start and expanded to within the detectable range during the period of study. This was even more striking for genes such as *SRSF2-P95H / U2AF1*, where 2/6 (33%) and 4/8 (50%) became detectable on-study, respectively.
- The shift in relative prevalence of CH driven by different gene mutations (Figure 1d, lower panel) offers further support to our finding that the impact of ageing on clonal behaviour differs by driver gene identity. Most notable here is the shift in relative prevalence of the two most common drivers of CH with advancing age, with *DNMT3A* dominating in earlier life and *TET2* dominating in late old age. A recent manuscript also identified a higher prevalence of *TET2*-mutant CH in a very old cohort, with *DNMT3A* more common in a slightly younger cohort [8]. Previous large studies of younger individuals also showed a striking dominance of *DNMT3A*-over *TET2*-mutant CH [9,10]. This again reflects the early initiation and age-related slow-down of *DNMT3A*-mutant clones, vs the more age-agnostic behaviour of *TET2*-mutant CH (Fig. 4i-j).

In summary, we provide additional evidence to corroborate findings from our time-series and lifelong phylogenetic analyses data that refute assumptions of lifelong constancy of clonal fitness (for many common mutations) and instead provide robust evidence that many clones grow at different rates during different stages of life.

We now include a comparison with the study of Watson *et al.* in our Discussion (Lines 383-388; Supplementary Table 11) and make reference to the recent publication mentioned above supporting our finding that *TET2* becomes the dominant driver of CH in advanced old age (reference number 43 in the main manuscript).

- Is their analysis in Figure 5a controlled for starting clone size?

This is an interesting question, especially given the association we previously reported between an individual’s risk of developing AML and the size of their largest driver clone [2]. In fact our analysis was not dependent on clone size, as clone size is by definition a derivative of growth rate and time. i.e. if sampled later, most clones would be bigger. However, unlike clone size, growth rate is stable over long periods and here we attempted to isolate the impact of this on AML risk. To make our comparisons valid, we actually compared growth rate to gene-level AML risks (agnostic of clone size) from our previous paper.

In contrast, in the Abelson *et al* study, we showed that clones destined to become AML within a few years (median 7-8 years) had actually already advanced to a bigger size. So, whilst clone size is a predictor of AML risk in this context, this is probably because it “captures” a clone that is closer to leukaemia. In other words, if we had samples from 20-30 years prior to AML development, then

clone size would probably not have discriminated between those who did and those who did not develop AML.

The limited value of starting (or other) clone size to our comparisons can be demonstrated by examining the relationship between average growth rate and median VAF at study entry vs study exit, which changes with time/age (Figure R6), unlike growth rate that remained stable over long periods. This illustrates why any single time point VAF measure in a cohort of individuals is not a universal quantity as it depends on the age of the individual. For this reason, it would not be appropriate to use clone size in our comparison with leukaemia risk. However, we note that, unlike in a cohort, having a large clone may be an indicator of being at greater risk of leukaemia development for a given individual as this suggests both faster growth and a more advanced stage.

Figure R6. Associations between CH driver gene growth per year and: (A) median VAF at study entry; (B) median VAF at study exit.

We now highlight the fact that our analysis is agnostic of clone size in our manuscript (Line 643).

Referee #3 (Remarks to the Author):

Primary uniqueness of this article is the collection of temporal variant frequencies in 56 genes collected over 13 years in 385 older individuals. This allowed for a study of longitudinal dynamics of clonal haematopoiesis (CH), with a few observations made and inferences drawn. Because the individuals were relatively old, some variants seeded lineages that grow faster than others (CH), including those in some key genes involved in CH-related cancers. Overall, however, I found this manuscript to be mostly descriptive with no scintillating discovery. The analysis of generic variation observed, while competent, is light on novelty.

Overall, this work is more suitable for a specialist human genetics journal. I have a few other comments, which are given below.

We thank reviewer #3 for outlining our findings and for assessing our manuscript and note that, unlike reviewers #1 and #2, we have not succeeded in convincing this reviewer about the profound nature of our discoveries.

In a nutshell, our study gives fundamental new insights into the lifelong dynamics of a common somatic phenomenon, that revise our understanding of its natural history and also, by extension, its relevance to human disease. For example, the discovery that CH clones expand at a predictable rate in old age, that is primarily influenced by driver mutation identity, has direct implications for newly established programmes for prevention of myeloid malignancies. Also, the discovery of the different lifelong behaviours of the three main subtypes of CH (driven by mutations in *DNMT3A*, *TET2* and splicing genes) forces a wholesale rethink of the cell-intrinsic and cell-extrinsic factors involved in clonal selection, and proposes that these almost certainly differ between CH subtypes. It also provides a convincing explanation for the changing prevalence of different CH subtypes with age (also highlighted by Reviewer #2). Furthermore, the discovery of widespread clonal deceleration (with notable exceptions) with advancing age, in the context of increasing oligoclonality, is a fundamental insight into the impact of ageing on haematopoiesis, with implications across the whole of haematology/immunology.

For example, *DNMT3A* is the most common CH driver gene but it has only limited leukemogenic potential. Previous studies using cross-sectional data deduced that *DNMT3A* mutations impart high fitness advantage [7], but while this is true when fitness is averaged over the earlier part of life, our data show that this high fitness is transient and clonal growth has decelerated by old age. In contrast, we show that mutations in other genes, such as *SRSF2* and *U2AF1*, exhibit rapid clonal growth specifically in old age, which provides a very plausible explanation for the sharp rise in prevalence of myeloid malignancies driven by these gene mutations in old age.

Such observations are profound as they touch upon the basic mechanisms governing somatic clonal evolution in a tissue that allows its serial study over long periods of time, informing how they relate to leukaemogenesis and ageing. Lessons learned in this way from the study of CH are likely to have parallels in other tissues, and can help to investigate the equivalent phenomena therein.

Methodologically, our study is the first to measure clonal dynamics in a primary tissue over a sustained period of more than a decade. We focused on older individuals because of the higher prevalence of CH in old age, but augmented our longitudinal study by phylogenetic analyses, which provide novel insights into the dynamics of CH clonal foundations. The phylodynamic analysis not only confirmed the clonal behaviours inferred by our longitudinal data, but also revealed the presence of coexisting clonal expansions without known drivers, which was discussed in greater detail in our accompanying manuscript by Mitchell et al, which was warmly received.

In the following, we provide detailed answers to the more specific questions raised by the reviewer. We have conducted a series of supplementary analyses to address all comments and to further strengthen the veracity and importance of our findings.

(1) Authors do not discuss any functional genomics aspects of variants. For example, truncating and missense mutations show similar patterns and they are all heterozygotes. Why are the patterns the same? What about their haplosufficiency, dominance, and loss/gain-of-function?

We agree that when we compare the impact of missense and truncating mutations on clonal growth dynamics, we see no significant difference for the two most common driver genes, *TET2* and *DNMT3A* (Extended Data Fig. 3j). While we do not investigate the reasons for this in our manuscript, there is recent evidence that the functional consequences of these two types of mutation can be similar in *DNMT3A*, with many missense mutations leading to protein instability and degradation [11], as truncating mutations do. Also, for *TET2*, whilst truncating mutations are spread fairly evenly across the gene, missense mutations are clustered in its conserved domains (Extended Data Fig. 2 and [12]), in keeping with both types of mutation resulting in reduced *TET2* function.

In comparison, for *TP53* and *CBL* we found that truncating and missense mutations had distinct impacts on clonal growth (missense faster in *TP53*, truncating faster in *CBL*; Extended Data Fig. 3j). It is interesting to note the previously reported functional differences in *TP53* mutations, with missense variants exerting a strong dominant negative effect [13], which would be predicted to be more detrimental than heterozygous truncating mutations, and compatible with a stronger phenotype (in our study, faster expansion). It is also notable that prognosis in some haematological malignancies is worse in individuals with missense vs truncating *TP53* mutations and it is plausible that this relates to a stronger anti-apoptotic phenotype associated with the former [14,15].

With regard to the haplosufficiency/dominance of CH mutations, it is clear that heterozygous loss-of-function mutations (eg. those in *DNMT3A* and *TET2*) are able to drive clonal expansions. However, it is also true that some mutations operate in a dominant-negative manner, for example *TP53* substitutions. There are also specific examples of gain-of-function mutations, such as substitutions in *IDH1/2*, *JAK2* and splicing genes, and truncating mutations in *PPM1D*, which are associated with increased PPM1D protein levels.

Despite the fact that CH driver mutations are almost always heterozygous, there is evidence that bi-allelic mutations can have a stronger phenotype [16]. However, this phenomenon is rare and seen in only ~2% of CH driver mutations in a recent study that combined detection of both

substitutions/indels and copy number changes [17]. Also, for some driver mutations, homozygosity is detrimental or even lethal to cells, as in the case of splicing gene mutations [18–20].

Our study did not intend to investigate the mechanisms by which CH driver mutations confer fitness advantage, but instead aimed to characterise their impact on clonal growth behaviours. Nevertheless, we appreciate the reviewer's point and do now allude to some of the functional aspects in our manuscript and make reference to relevant published mechanistic studies pertaining to *TP53* and *DNMT3A*, as examples of genes in which substitutions and truncating mutations have different (*TP53*) vs similar (*DNMT3A*) mechanistic consequences (Lines 163-164, 166-167).

(2) Page 15-16, lines 448-449: The assumption of the VAF to be representative of the prevalence of a single clone needs to be re-evaluated. For example, in Figure 3b, clone 3 with Unknown driver 2 produced clone 2 with U2AF1 mutation. This means that the VAF of Unknown driver 2 represents clones 3 and 2. The impact of this violation on the conclusion should be discussed, unless I am mistaken.

This is a good point and we thank the reviewer for highlighting it. To address it, we first considered the possible scenarios for any two given driver mutations (A and B) detected in a single individual:

- **Mutually exclusive:** A and B exist in separate clones and each is the sole driver of the expansion of its cognate clone;
- **Sub-clonal:** both A and B appear in a previously expanded clone (the previous expansion having been driven by mutation X, an unknown driver) but each is acquired in a separate cell and drives a new expansion. As such, each subclone can be characterized separately as *clone X acquiring mutation A* and *clone X acquiring mutation B* (X->XA and X->XB, respectively);
- **Co-clonal:** both A and B are in the same clone (nominally A was acquired first, followed by B (A->AB)).

There is now evidence from single-cell mutational profiling that, unlike in Acute Myeloid Leukaemia (AML), **mutual exclusivity** is the most common scenario in CH, with only rare instances of CH driver mutations co-occurring in the same cell [21]. Indeed, sequencing of single-cell derived colonies from our phylogenetic studies (Figure 3) and those of Mitchell et al (our companion manuscript [5]), confirm that *known* CH driver mutations usually drive expansion of clones in the absence of other *known* CH driver mutations - i.e. CH drivers do not commonly co-occur in the same cell. Our additional analyses of clonality (detailed in response to the following Review point (3) and included in our new Supplementary Note 3) support this.

If it were the case that two CH driver mutations were present in the same clone, and each influenced overall clone fitness, we would expect the clone to grow at a rate different from that predicted by the identity of either of the two mutations alone. In our longitudinal model of clonal dynamics, this would be captured in the 'unknown-cause' growth effect, i.e. the component of growth not explained by the driver mutation in question. Importantly, however, we see no association between the number of CH driver mutations present within a single individual and this unknown-cause effect, consistent with a paucity/absence of co-operating CH drivers (Supplementary Note 3, Figure S8, left panel). In addition, if there was sequential acquisition of CH drivers within the same cell, each

contributing to clonal fitness, we would expect step-changes in clonal growth rate over time. However, in the vast majority of cases (92.4%), clones grew at a constant rate during the period of study. Furthermore, the number of driver mutations per individual was not associated with the fraction of clones growing at a fixed rate (Supplementary Note 3, Figure S8, right panel).

Therefore, it is reasonable to assume that known CH driver mutations do not commonly co-occur in the same cell, and that a mutation's VAF represents the size and trajectory of its associated clone. Also, even if there is occasional co-clonality, this does not affect our overall inferences/conclusions. **We have reworded our manuscript to reflect this (Lines 464-465).**

With regard to potential interactions between known and unknown drivers of clonal expansions, our phylogenies and those of Mitchell et al [5] demonstrate instances of known CH drivers being acquired in clones already expanded by unknown drivers, such as the *U2AF1*-mutant clone in PD34493 (Figure 3b) referred to by the Reviewer. In this particular case, the *U2AF1* mutation does contribute to the dynamics of the 'unknown driver 2' clade; the average annual growth of the clade is estimated to be 24.3% vs 16.9% if we include vs exclude the *U2AF1*-mutant expansion from the trajectory analysis, respectively. While we cannot directly quantify the impact of unknown CH drivers on the growth dynamics associated with known CH drivers, we know that any such influence would be captured in the 'unknown-cause' effect in our longitudinal model. In this model, we find that a clone's growth rate is determined predominantly by the identity of its known CH driver mutation and the unknown-cause effect is usually comparatively small, so any influence of unknown drivers on overall driver growth dynamics must also be minor.

(3) Also, line 448 mentions that phasing different mutations into specific clones was impossible. However, there are many computational methods to phase different mutations into specific clones when VAF is available from multiple timepoints or samples (e.g., PyClone and CITUP). Many studies with deep sequencing have reported mutation clusters. Would that information be not helpful to even reconstruct better mutational history?

We agree that there are several methodologies for phasing clonal/subclonal mutations identified in deep sequencing data. However, these rely on the identification of a sufficient number of somatic mutations to allow for this to be performed in a statistically robust manner. In the longitudinal part of our study, we performed targeted sequencing of 56 CH-associated genes in 1593 samples with emphasis on the robust identification of CH driver mutations, even when these were present in small clones. The fact that, for good reasons, we focused on a small part of the genome (56 genes) meant that we only identified a small number of somatic mutations in each individual, limiting the ability to phase drivers into different clones, despite the availability of VAF measurements from multiple timepoints. Also, importantly, we find that the number of mutations within an individual had no impact on inferred clonal growth rates; i.e. driver-specific growth rates observed in individuals with single mutations (where co-clonality cannot exist), did not differ from those observed in individuals with multiple mutations (where co-clonality could theoretically exist) (Supplementary Note 3, Fig. S8).

Despite the unsuitability of our data for use in this context, we now attempt to infer the clonality of driver mutations using four different approaches, and show these analyses in a new supplementary note (Supplementary Note 3).

Two of these approaches gave results that were not informative for phasing mutations. *First*, PyClone proposed co-clonality in *all* individuals with more than one mutation, which is implausible because it (i) conflicts with the available literature on clonal architecture in CH, where a single driver per CH clone is the norm [21], and (ii) is functionally implausible for certain mutation pairs (see Supplementary Note 3). *Second*, we tried to use clonal growth rates to inform clonal structure, assuming that mutations in a single individual could exist in the same clone if their VAFs were changing at a similar rate over time. For this, we calculate a background distribution of differences in growth rates between any two randomly picked mutations from our cohort (across individuals) and find that these differences in mutation-specific growth rates *between* individuals were indistinguishable from differences in growth rates *within* individuals (Supplementary Note 3, Fig. S7), rendering this approach also uninformative.

Two other approaches for inferring clonal structure gave some tentative information, albeit inconclusive. *First*, we assumed that mutations in a single individual could exist in the same clone if their VAFs were of a similar size at all time-points; this was the case in 13.6% of mutation pairs. However, this does not prove co-clonality, as it is also possible that the mutations exist in distinct clones with similar size trajectories. *Second*, using the pigeonhole principle, which stipulates that heterozygous mutations can only exist in entirely separate clones if their VAFs sum to ≤ 0.5 , we identify only 6 pairs of mutations which could be co-clonal.

In summary:

1. While knowing the clonal architecture could be informative, we have no way to reliably resolve this given the very targeted nature of our sequencing data.
2. The actual number of co-clonal mutations is likely to be very low given the reported rarity in CH [21].
3. Most importantly, the number of mutations within an individual had no significant impact on our clonal growth rate inferences (Supplementary Note 3, Fig. S8), suggesting that even if some co-clonal events did exist, the main findings of our study would not change.

(4) For the single cell sequencing data, it is unclear why MPBoot was sufficient. This regular approach is not going to be effective if there are many missing bases and erroneous mutation assignments. There is a critical need to provide information on data quality and the robustness of the inferences to the choice of method (e.g., MP vs SCITE).

Our data is not 'single-cell' data, as it was generated by *in vitro* expansion of single cells to colonies, each containing hundreds to thousands of cells, followed by standard Illumina short-read sequencing (WGS) to a mean depth of 15X per sample. We apologise if we had not made this sufficiently clear and have now included additional text in our manuscript to make the distinction between our data type and single-cell data (Lines 195-196 and 560). Our method resulted in confident genotyping at

the vast majority of loci used for tree-building. For each of the three phylogenies (PD34493, PD41276, PD41305), the percentage of missing genotypes going into tree-building was only 1.5%, 1.4% and 1.3% respectively (see below). The same methodology was used by Mitchell et al, including for the four phylogenetic trees used in our manuscript [5].

Nevertheless, we agree that it is important for the reader to get a better feel for the cleanliness of the data, and therefore have updated the manuscript by: (i) stating the proportion of missing genotypes per individual (Lines 592-594) and (ii) providing genotyping heatmaps for each individual across the mutations used for tree building (Extended Data Fig. 5a-c). Additionally, we assessed the internal consistency of shared mutation data using the disagreement score, which demonstrates that the vast majority of allocated genotypes were concordant with phylogenetic assumptions, with ~100,000-fold lower scores for the data compared to random shuffles of the genotypes at each locus. We now include detail on this in a new Supplementary Methods section (Lines 1018-1032) and new Extended Data Figure 5d.

Given the high quality of our data, we feel that MPBoot is an appropriate method for phylogeny inference for our data. We do agree however that SCITE is also an appropriate method. For this reason, we have now done a comparison of the two approaches, to check that they are broadly concordant. This is indeed the case; SCITE produced phylogenies with high agreement to the original MPBoot phylogenies, with Robinson-Foulds similarities of 0.989 (PD34493), 0.996 (PD41305) and 1.000 (PD41276). We now show this comparison in Supplementary Methods (Lines 1033-1042) and new Extended Data Fig. 5e.

(5) The accuracy of the inferred phylogenetic tree from single cell sequencing data are likely tentative, as the MP method cannot deal with data containing many missing data and incorrect base assignments. Methods designed for low-quality and single-cell sequencing data should be used (e.g., SCITE).

As discussed above, our data was not derived from single cells and does not contain many missing genotypes or incorrect base assignments (Extended Data Fig. 5a-c). Therefore, we feel that maximum parsimony (using MPBoot) is a reasonable approach, with results comparable to SCITE (Extended Data Fig. 5e).

(6) In fact, the quality of sequence data should be reported, e.g., rate of unknown base assignments and sensitivity of variant calling (page 19 lines 579-580).

We agree that the rate of unknown base assignments is important for the reader to assess the quality of the data. As discussed above, we have therefore provided a new figure with a heatmap of the genotype matrix inputted into the tree-building algorithm for each phylogeny (Extended Data Fig. 5a-c). We have also included the exact proportion of missing genotypes in the text of the Methods (Lines 592-594).

The sensitivity of variant-calling was calculated based on the sensitivity of our variant-calling algorithm to call germline SNPs in each sample, as described previously [22]. Mean sensitivity of variant-calling for each of the three phylogenies was 85.4%, 87.0% and 83.5% for PD41305, PD41276 and PD34493, respectively. We also agree that these are important parameters for the reader to assess, and we now include sensitivity values in our manuscript (Lines 601-604).

(7) Page 19 lines 582-584: The time axis was scaled linearly from the conception (the root of the tree) to the age of sampling (the tips), and constant mutation rates were assumed to generate a timeline. It is unlikely that mutation rate was constant from the time of the conception to the age of sampling. In contrast, there may be many phases of clock ticking at different rates.

The reviewer raises an important point relating to the relationship between molecular time on the phylogenetic tree axis and chronological age. There is now a growing body of evidence that human haematopoietic stem cells (HSCs) accumulate somatic mutations at a constant rate (~17 mutations per year), both within and between individuals, throughout ex-utero life [5,23–26]. However, recent evidence suggests that the rate is higher *in utero*, with ~55 mutations acquired during this first period of life [5,22]. In addition, refuting the long-held assumption that DNA replication and cell division are the major sources of somatic mutation acquisition, recent evidence suggests that the major determinant of HSC mutation rate is, instead, time [23,26]. This means that changes in HSC generation time during life, or in response to transient perturbations, will have minimal impact on mutation burden.

We therefore conclude that the assumption of a linear accumulation of mutations during *ex utero* life – the relevant timescale for our inference – is reasonable. Also, we now account for the higher rate of mutation acquisition *in utero* on our phylogeny time axes, by assigning the first 55 mutations to this period, scaling the remaining mutation time linearly by age, and adjusting downstream analyses accordingly (Figure 3). We have updated the Methods section to reflect this adjustment to scaling (Lines 607-612). Since these 55 mutations constitute only a very small fraction of the total number of mutations acquired over the lifetimes of the elderly individuals we studied, these adjustments made only marginal differences to estimates of ages at clonal onset and expansion rates, and have no impact on our conclusions.

(8) Did the dN/dS ratio the same across time in variation data and in sc-Seq data, excluding the CH clones?

The idea that selection strength might change with time is very interesting. Indeed, our own data suggest that selection strength changes with time, and that the relationship between age and ‘fitness’ is gene-specific. For example, we found that *DNMT3A*-mutant clones preferentially expanded early in life and displayed slower growth in old age (consistent with *diminishing* selective pressure with advancing age), while splicing gene mutations only drove expansion later in life (consistent with *increasing* selective pressure with advancing age).

In response to this comment, we take ‘variation data’ to be our longitudinal targeted sequencing data, and ‘sc-Seq data’ to be our haematopoietic colony-derived whole genome sequencing data. As requested, we now quantify changes in dN/dS ratio over time in both sets of data. Selection can be defined as the rate of reaching a particular frequency or fixation in the population, and the ratio dN/dS is an attempt to quantify the effect of selection in relation to the rate of neutral drift. We use the dNdScv algorithm, an implementation of dN/dS that corrects for trinucleotide mutation rates, sequence composition, and variable mutation rates across genes [27].

From the 385 individuals included in our longitudinal data set, we now compare the dN/dS ratio at the time of study entry (median age 69.3 years) with the ratio at the end-of-study (median age 81.3 years). We derive both global dN/dS ratios (Figure R7) and gene-specific dN/dS ratios (Figure R8).

Figure R7. Global dN/dS ratios across the targeted genomic regions at the start vs end of study.

Figure R8. Gene dN/dS ratios at the start vs end of study, for (A) missense and (B) truncating mutations.

While confidence intervals are too broad to make definitive conclusions, the trend is consistently for higher dN/dS ratios with advancing age. One interpretation of this is that selection tends to strengthen with age. However, higher dN/dS ratios at a later time-point might also reflect the fact that mutant clones have had longer to reach detectable levels. In this case, drivers would not be

'fitter' in older people, they would simply have had longer to impart their growth advantage and make their cognate clones detectable. For this reason, dN/dS ratios are imperfect measures of selection strength at a specific point in time, and instead reflect cumulative selection prior to that time. This highlights a strong advantage of our longitudinal data in allowing us to directly observe clonal growth and calculate fitness coefficients as more faithful measures of selection strength.

In order to use our 3 haematopoietic phylogenies to address the question of whether dN/dS ratios changed across time, we needed to compare mutations acquired early with those acquired later. Since mutations can be 'timed' only in relation to whether they were present before or after a branching event, we elected to define 'early' mutations as those on shared branches (coloured red on the example phylogeny in Figure R9), and 'late' mutations as those on private branches (coloured grey in Figure R9).

Figure R9. Example phylogeny (PD41305), illustrating our definition of 'early' and 'late' mutations, according to whether they lie on shared (red) or private (grey) phylogeny branches.

We combined the phylogenies into a single analysis to maximise our power to detect any signal from the limited number of coding mutations across the genomes of just 3 individuals. Taking age to be the midpoint of a branch, the mean age of 'early' mutations (along shared branches) was 35.6 years, and the mean age of 'late' mutations (along private branches) was 59.2 years. There were 245 and 1885 coding somatic mutations in the 'early' and 'late' groups, respectively. Applying the dNdScv algorithm, we observe a trend towards higher global dN/dS ratios along shared vs private branches (Figure R10).

Figure R10. Global dN/dS ratios among shared vs private mutations.

Excluding known CH driver mutations (defined as those in the 17 genes included in our longitudinal model), or excluding any mutation in a cancer gene (defined as those in Tier 1 of the Cancer Gene Census; <https://cancer.sanger.ac.uk/census>), made almost no impact on the dN/dS ratios derived as above.

Since mutations acquired along shared phylogeny branches are, by definition, those preceding clonal expansion, it is expected that selection would be strongest here, as compared to along private branches, where mutations have not, by definition, instigated clonal expansion. This is despite shared branches occurring at younger ages than private branches. This inability to disentangle age from clonal expansion limits the utility of phylogenies for investigation of changes in selection with time.

In summary, while the dN/dS ratio can be an important tool for measuring selection, its utility for assessing changes in selection with age is limited, at least when applied to longitudinal VAF data or phylogenetic data, for the reasons discussed above. By contrast, the analyses of clonal growth dynamics in our manuscript allow us to derive driver-specific fitness effects, without being subject to these limitations.

Nevertheless, the dN/dS analyses we performed in response to this insightful comment are informative in their own right and we now reflect this in our manuscript by including new text in the main section (Lines 444-446 and 458-461) and adding a new Supplementary Note (Supplementary Note 2).

References

1. Sant M, Allemani C, Tereanu C, De Angelis R, Capocaccia R, Visser O, et al. Incidence of hematologic malignancies in Europe by morphologic subtype: results of the HAEMACARE project. *Blood*. 2010;116: 3724–3734.
2. Abelson S, Collord G, Ng SWK, Weissbrod O, Mendelson Cohen N, Niemeyer E, et al. Prediction of acute myeloid leukaemia risk in healthy individuals. *Nature*. 2018;559: 400–404.
3. Therneau TM. A Package for Survival Analysis in R. 2020. Available: <https://cran.r-project.org/package=survival>
4. McKerrell T, Park N, Moreno T, Grove CS, Ponstingl H, Stephens J, et al. Leukemia-associated somatic mutations drive distinct patterns of age-related clonal hemopoiesis. *Cell Rep*. 2015;10: 1239–1245.
5. Mitchell E, Chapman MS, Williams N, Dawson K, Mende N, Calderbank EF, et al. Clonal dynamics of haematopoiesis across the human lifespan. *bioRxiv*. 2021. p. 2021.08.16.456475. doi:10.1101/2021.08.16.456475
6. Watson CJ, Papula A, Poon YPG, Wong WH, Young AL, Druley TE, et al. The evolutionary dynamics and fitness landscape of clonal haematopoiesis. *bioRxiv*. 2019; 569566.
7. Watson CJ, Papula AL, Poon GYP, Wong WH, Young AL, Druley TE, et al. The evolutionary dynamics and fitness landscape of clonal hematopoiesis. *Science*. 2020. doi:10.1126/science.aay9333
8. Rossi M, Meggendorfer M, Zampini M, Tettamanti M, Riva E, Travaglino E, et al. Clinical relevance of clonal hematopoiesis in the oldest-old population. *Blood*. 2021. doi:10.1182/blood.2021011320
9. Jaiswal S, Fontanillas P, Flannick J, Manning A, Grauman PV, Mar BG, et al. Age-related clonal hematopoiesis associated with adverse outcomes. *N Engl J Med*. 2014;371: 2488–2498.
10. Genovese G, Köhler AK, Handsaker RE, Lindberg J, Rose SA, Bakhoum SF, et al. Clonal hematopoiesis and blood-cancer risk inferred from blood DNA sequence. *N Engl J Med*. 2014;371: 2477–2487.
11. Huang Y-H, Chen C-W, Sundaramurthy V, Slabicki M, Hao D, Watson CJ, et al. Systematic profiling of DNMT3A variants reveals protein instability mediated by the DCAF8 E3 ubiquitin ligase adaptor. *Cancer Discov*. 2021. doi:10.1158/2159-8290.CD-21-0560
12. Ferrone CK, Blydt-Hansen M, Rauh MJ. Age-Associated TET2 Mutations: Common Drivers of Myeloid Dysfunction, Cancer and Cardiovascular Disease. *Int J Mol Sci*. 2020;21. doi:10.3390/ijms21020626
13. Boettcher S, Miller PG, Sharma R, McConkey M, Leventhal M, Krivtsov AV, et al. A dominant-negative effect drives selection of TP53 missense mutations in myeloid malignancies. *Science*. 2019;365: 599–604.
14. Eskelund CW, Dahl C, Hansen JW, Westman M, Kolstad A, Pedersen LB, et al. TP53 mutations identify younger mantle cell lymphoma patients who do not benefit from intensive chemoimmunotherapy. *Blood*. 2017;130: 1903–1910.

15. Rodrigues JM, Hassan M, Freiburghaus C, Eskelund CW, Geisler C, Rätty R, et al. p53 is associated with high-risk and pinpoints TP53 missense mutations in mantle cell lymphoma. *Br J Haematol.* 2020;191: 796–805.
16. Wang L, Wheeler DA, Prchal JT. Acquired uniparental disomy of chromosome 9p in hematologic malignancies. *Exp Hematol.* 2016;44: 644–652.
17. Saiki R, Momozawa Y, Nannya Y, Nakagawa MM, Ochi Y, Yoshizato T, et al. Combined landscape of single-nucleotide variants and copy number alterations in clonal hematopoiesis. *Nat Med.* 2021;27: 1239–1249.
18. Lee SC-W, North K, Kim E, Jang E, Obeng E, Lu SX, et al. Synthetic Lethal and Convergent Biological Effects of Cancer-Associated Spliceosomal Gene Mutations. *Cancer Cell.* 2018;34: 225–241.e8.
19. Wadugu BA, Nonavinkere Srivatsan S, Heard A, Alberti MO, Ndonwi M, Liu J, et al. U2af1 is a haplo-essential gene required for hematopoietic cancer cell survival in mice. *J Clin Invest.* 2021;131. doi:10.1172/JCI141401
20. Fei DL, Motowski H, Chatrikhi R, Prasad S, Yu J, Gao S, et al. Wild-Type U2AF1 Antagonizes the Splicing Program Characteristic of U2AF1-Mutant Tumors and Is Required for Cell Survival. *PLoS Genet.* 2016;12: e1006384.
21. Miles LA, Bowman RL, Merlinsky TR, Csete IS, Ooi AT, Durruthy-Durruthy R, et al. Single-cell mutation analysis of clonal evolution in myeloid malignancies. *Nature.* 2020;587: 477–482.
22. Spencer Chapman M, Ranzoni AM, Myers B, Williams N, Coorens THH, Mitchell E, et al. Lineage tracing of human development through somatic mutations. *Nature.* 2021;595: 85–90.
23. Abascal F, Harvey LMR, Mitchell E, Lawson ARJ, Lensing SV, Ellis P, et al. Single-molecule mutation detection unravels the mutational landscapes of differentiated cells. *Nature.* 2021.
24. Lee-Six H, Øbro NF, Shepherd MS, Grossmann S, Dawson K, Belmonte M, et al. Population dynamics of normal human blood inferred from somatic mutations. *Nature.* 2018;561: 473–478.
25. Osorio FG, Huber AR, Oka R, Verheul M, Patel SH. Somatic mutations reveal lineage relationships and age-related mutagenesis in human hematopoiesis. *Cell Rep.* 2018. Available: <https://www.sciencedirect.com/science/article/pii/S2211124718317601>
26. de Kanter JK, Peci F, Bertrums E, Rosendahl Huber A, van Leeuwen A, van Roosmalen MJ, et al. Antiviral treatment causes a unique mutational signature in cancers of transplantation recipients. *Cell Stem Cell.* 2021;28: 1726–1739.e6.
27. Martincorena I, Raine KM, Gerstung M, Dawson KJ, Haase K, Van Loo P, et al. Universal Patterns of Selection in Cancer and Somatic Tissues. *Cell.* 2017;171: 1029–1041.e21.

Reviewer Reports on the First Revision:

Referees' comments:

Referee #2 (Remarks to the Author):

The authors have done an excellent job responding to critiques from the reviewers.

Referee #3 (Remarks to the Author):

Authors have done a great job of carefully considering my comments (reviewer 3) as well as other reviewers' suggestions. I have no further concerns.